# LEARNING NEURAL ACOUSTIC FIELDS

## ABSTRACT

Our sensory perception of the world is rich and multimodal. When we walk into a cathedral, acoustics as much as appearance inform us of the sanctuary's wide open space. Similarly, when we drop a wineglass, the sound immediately informs us as to whether it has shattered or not. In this vein, while recent advances in learned implicit functions have led to increasingly higher quality representations of the visual world, there have not been commensurate advances in learning auditory representations. To address this gap, we introduce Neural Acoustic Fields (NAFs), an implicit representation that captures how sounds propagate in a physical scene. By modeling the acoustic properties of the scene as a linear time-invariant system, NAFs continuously map all emitter and listener location pairs to an impulse response function that can then be applied to new sounds. We demonstrate that NAFs capture environment reverberations of a scene with high fidelity and can predict sound propagation for novel locations. Leveraging the scene structure learned by NAFs, we also demonstrate improved cross-modal generation of novel views of the scene given sparse visual views. Finally, the continuous nature of NAFs enables potential downstream applications such as sound source localization. Qualitative results: sites.google.com/view/nafs-iclr-2022/.

## 1 INTRODUCTION

The sound of the ball leaving the bat, as much as its visible trajectory, tells us whether the hit is likely to be a home run or not. Our experience of the world around us is rich and multimodal, depending on integrated input from all of five sensory modalities – vision, touch, smell, hearing, and taste. Understanding underlying scene structure relies on a continuous and diverse set of inputs: the visual appearance and geometry of the physical world, the way sounds bound off of or are blocked by surfaces, the way odors diffuse across the local environment, and the way objects feel as we walk on them or touch them.

Recent progress in implicit neural representations has enabled the construction of continuous, differentiable representations of the visual world directly from raw image observations (Sitzmann et al., 2019; Mildenhall et al., 2020; Niemeyer et al., 2020; Yariv et al., 2020). These models typically utilize a neural renderer in combination with a learned implicit representation to jointly capture and render images of a scene. By leveraging the multiview consistent nature of visual observations, these methods can infer images of the same scene from novel viewpoints once trained on sparse views.

However, our perception of the physical world is informed not only by our visual observations, but also by the acoustics of environmental sounds. When we are in a large room, we expect to hear highly reverberant echos (e.g., as in a cathedral). Moreover, given only the acoustic properties of sounds in this space, we may infer a variety of other properties of the environment, for example, the location of the emitted sound or the underlying room layout. As a preliminary step in learning the acoustic properties of scenes, we explore modeling the underlying impulse response of audio reverberations.

Past work has explored capturing the underlying acoustics of a scene (Raghuvanshi & Snyder, 2014; 2018; Chaitanya et al., 2020). These models, however, require manually designing acoustic functions which, critically, prevent such approaches from being applied to arbitrary scenes. In this work, we extend this approach by constructing an implicit neural representation which captures, in a *generic manner*, the underlying acoustics of a scene. In particular, following (Raghuvanshi & Snyder, 2014), we define the acoustic modeling problem as modeling the impulse-response a listener receives given a sound emitted at an emitter location (as illustrated in Figure 1) and across all possible emitter-

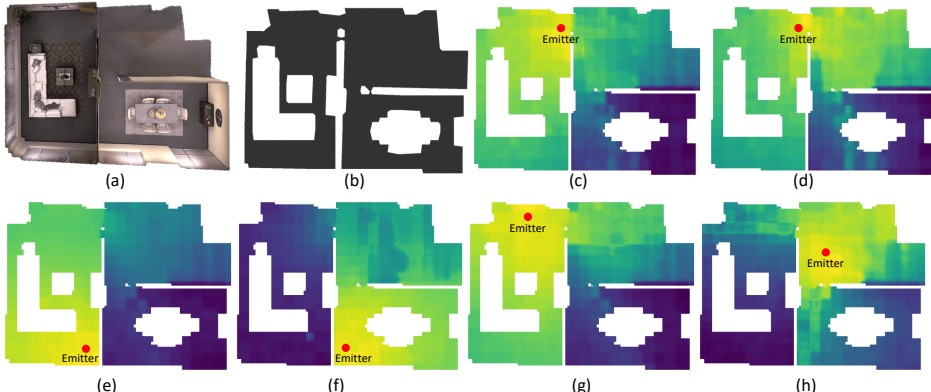

Figure 1: Our neural acoustic fields learn an implicit representation for acoustic propagation. **(a)** A 3D top-down view of the room. **(b)** Walkable regions shown in grey. **(c)** and **(d)** NAF captures the directional nature of sound. For a single emitter location, the left and right ears experience different loudness. **(e)-(h)** NAF constructs a continuous acoustic field is created as we move the sound emitter in the scene. For **(e)** and **(f)** note that the sound does not leak through the wall. For **(g)** and **(h)** note the portaling effect open doorways can have. Lighter colors indicate a stronger response. Response strength is determined by a log of summed STFT magnitude components for a fixed emitter location.

listener pairs in a scene. This learned representation inherently captures the pattern of all acoustic reverberations in a scene.

Learning a representation of scene acoustics poses several challenges compared to the visual setting. First, how do we represent, in high fidelity, the underlying impulse response at each emitter-listener position? While we may represent the visual appearance of a scene with an underlying three-dimensional vector, an acoustic reverberation (represented as an impulse response) can consist of over 20000 values and, thus, is significantly harder to capture. Second, how do we learn an acoustic neural representation that densely generalizes to novel emitter-listener locations? In the visual setting, ray-tracing can enforce view consistency across large portions of a visual scene (modulo occlusions). While in principal, in a similar manner, we may reflect acoustic "rays" in our scene to obtain an impulse response, a intractable number of rays are necessary to obtain the desired representation.

To address both challenges, we propose Neural Acoustic Fields (NAFs). To capture the complex signal representation of impulse responses, NAFs encode and represent an impulse-response in the Fourier frequency domain. Motivated by the strong influence of nearby geometry on anisotropic reflections (Raghuvanshi & Snyder, 2018), we propose to condition NAFs on local geometric information present at both the listener and emitter locations when decoding the impulse response. In our framework, local geometric information is learned directly from impulse responses. Such a decomposition facilitates the transfer of local information captured from training emitter-listener pairs to novel combinations of emitters and listeners

By modeling the dense acoustic fields of an environment, NAFs are a powerful method to extract structural information about the scene, which we demonstrate is useful across a variety of different tasks. In the cross-modal setting, we demonstrate how the learned acoustic structure can be utilized to aid learned visual representations, improving novel view synthesis. Further, by directly utilizing the acoustic information in NAFs, we demonstrate how one can localize different sound sources.

In summary, we present Neural Acoustic Fields (NAFs), a neural implicit representation which captures the underlying acoustics of a scene in a compact and spatially continuous fashion. We show that NAFs are able to outperform baselines in modeling scene acoustics, and provide detailed analysis of the design choices in NAFs. We further illustrate how the structure learned by NAFs, can improve cross-modal generation of novel visual views of a scene. Finally, we illustrate how the continuous nature of NAFs enable the downstream application of sound source localization.

## 2    RELATED WORK

**Audio Field Coding**    There is a rich history of encoding methods for 3D spatial audio. These approaches largely fall into two categories. The first approach encodes the sound field at a user-centric location by capturing the sound from spatially distributed sources (Gerzon, 1973; Breebaart

et al., 2005; Pulkki, 2007). While they may leverage perceptual cues that create the sense of spatial audio, they do not allow the listener to freely traverse the scene and experience sound from different locations. The second approach aims to model the sound heard as a listener moves in a scene (Raghuvanshi & Snyder, 2014; Mehra et al., 2014; Raghuvanshi & Snyder, 2018; Chaitanya et al., 2020). Since the complete acoustic field of a scene is computationally prohibitive to simulate in real time, and expensive to store in full fidelity, these methods have relied on a handcrafted encoding of the acoustic field, prioritizing efficiency above reproduction fidelity. Our work allows a listener to move and experience sounds that come from anywhere in a scene, and can represent the acoustic field continuously at high fidelity by directly learning from data.

**Implicit representation** Our approach towards modeling the underlying acoustics a scene relies on the use of a neural implicit representations. Implicit representations have emerged as a promising representation of 3D geometry Niemeyer et al. (2019); Chen & Zhang (2019); Park et al. (2019); Saito et al. (2019) and appearance Sitzmann et al. (2019); Mildenhall et al. (2020); Niemeyer et al. (2020); Yariv et al. (2020) of a scene. Compared to traditional discrete representations, implicit representations are a continuous mapping capable of capturing data at an "infinite resolution". Jiang et al. (2020) proposed a grid based representation for implicit scene reconstruction, while more recently DeVries et al. (2021) has adopted spatial conditioning for 3D image synthesis, where in both settings, the grid enables a higher-fidelity encoding of the scene. Our work also leverages local grids to model acoustics, but as an inductive bias and way to generalize to novel inputs.

**Audio-Visual Learning** Our work is also closely related to joint modeling of vision and audio. By leveraging the correspondence between vision and audio, work has been done to learn unsupervised video and audio representations (Aytar et al., 2016; Arandjelovic & Zisserman, 2017), localize objects that emit sound (Senocak et al., 2018; Zhao et al., 2018), and jointly use vision and audio for navigation (Chen et al., 2020). Recent work aims to propose plausible reverberations or sounds from image input (Singh et al., 2021; Du et al., 2021), these approaches model the phase-free log-magnitude STFT using either convolution or implicit functions, which we also utilize. Different from them, our work leverages the geometric features learned by modeling acoustic fields to improve the learning of 3D view generation.

## 3 METHODS

We are interested in learning a generic acoustic representation of an arbitrary scene, which can capture the underlying sound propagation of arbitrary sound sources across both seen and unseen locations in a scene. We first review relevant background information towards modeling environment reverberations. We then describe Neural Acoustic Fields (NAFs), a neural field which we show can capture, in a generic manner, the acoustics of arbitrary scenes. We further discuss how we can parameterize NAF in a manner so that it can capture acoustics property even at unseen sound sources and listener positions. Finally, we discuss the underlying implementation details of our model.

### 3.1 BACKGROUND ON ENVIRONMENTAL REVERBERATION

The sound emitted by a sound source undergoes decay, occlusion, and scattering due to both the geometric and material properties of a scene. For a fixed location pair $(q, q')$, we define the impulse-response at a listener position $q$, as the sound pressure $p(t; q, q')$ induced by an impulse at $q'$. Such behavior can be concisely and elegantly modeled utilizing the linear wave equation (Pierce, 2019):

$$\left[ \frac{1}{c^2} \frac{\partial^2}{\partial t^2} - \nabla^2 \right] p(t, q, q') = \delta(t)\delta(q - q'), \tag{1}$$

where $c$ is the speed of sound, $p$ is the sound pressure, $(q, q')$ being the listener and emitter location respectively, and $\delta$ the Dirac delta representing the forcing function, where we refer to sound pressure $p(t; q, q')$ as the impulse-response observed at listener position $q$.

Given an accurate model of the impulse-response $p(t; q, q')$ described in Eqn (1), we may model audio reverberation of any sound waveform $s(t)$ emitted at $q'$, by computing the response $r(t, q, q')$ at listener location $q$ by querying the continuous field and using temporal convolution:

$$r(t; q, q') = s(t) \circledast p(t; q, q') \tag{2}$$

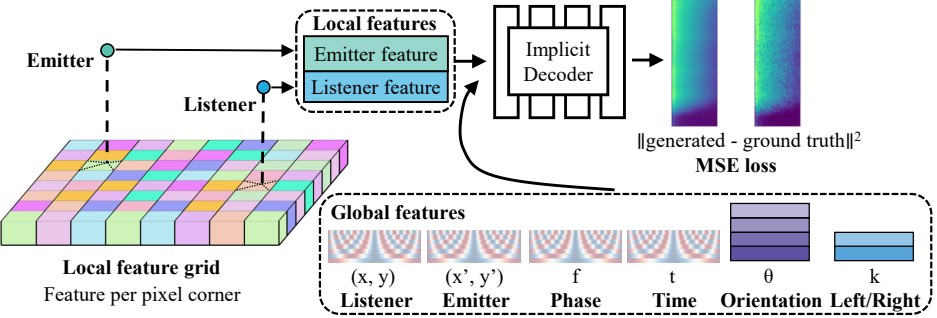

Figure 2: Overview of NAF. Given a listener position and an emitter location, we first query a grid for local features using bilinear interpolation. We compute the sinusoidal embedding of the positions, phase, and time, and query a discrete embedding matrix using the orientation and left/right ear. These features are fed to an implicit decoder. Our method is trained with a MSE loss with impulse responses.

## 3.2 NEURAL ACOUSTIC FIELDS

We are interested in constructing a continuous representation of the underlying acoustics of a scene, which may specify the reverberation patterns of an arbitrary sound source. The parameterization of an impulse-response introduced in Section 3.1 provides us with a method to model audio propagation when given an omnidirectional listener and emitter. To construct a model of a directional listener, we further model the 3D head orientation $\theta \in \mathbb{R}^2$, and ear $k \in \{0, 1\}$ (binary left or right) of a listener, in addition to the spatial position $\boldsymbol{q} \in \mathbb{R}^3$ of the listener and $\boldsymbol{q}' \in \mathbb{R}^3$ of the emitter.

We may then model the time domain impulse response $\boldsymbol{v}$ using a neural field $\Phi$ which takes as input the listener and emitter parameters:

$$\Phi : \mathbb{R}^8 \times \{0, 1\} \to \mathbb{R}^T, \quad (\boldsymbol{q}, \theta, k, \boldsymbol{q}') \to \Phi(\boldsymbol{q}, \theta, k, \boldsymbol{q}') = \boldsymbol{v} \qquad (3)$$

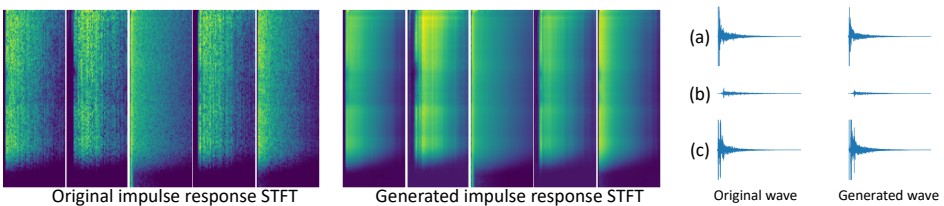

Figure 3: **Qualitative Visualization of Test Set Impulse Response Prediction.** Left: Example log-STFT of impulse responses and predictions from NAF. Right: **(a)-(c)** Three examples of ground truth waveform of the impulse response, and the corresponding NAF generated waveform reconstructed with Griffin-Lim.

Directly outputting the impulse-response $\boldsymbol{v} \in \mathbb{R}^T$ with a neural network is difficult to its high dimensional (over 20000 elements) and chaotic nature. A naïve solution would be further add $t$ as an additional argument our neural field, but we found that such a solution worked poorly, due to the highly non-smooth representation of the waveform (see Table A2). We instead encode the impulse-response utilizing a short-time Fourier transform (STFT) to create a log-magnitude spectrogram denoted $v_{\text{STFT}}$, which we find to be significantly more amenable to neural network prediction, due to the smooth nature of the frequency space. In Figure 3 we show spectrograms for ground truth impulse responses and those learned by our network.

Thus, our parameterization of NAF is a neural field $\Omega$ that is trained to estimate the impulse response function $\phi$, and outputs $\boldsymbol{v}_{\text{STFT}}$ for a given time and frequency coordinate:

$$\Omega : \mathbb{R}^{10} \times \{0, 1\} \to \mathbb{R}, \quad (\boldsymbol{q}, \theta, k, \boldsymbol{q}', t, f) \to \Omega(\boldsymbol{q}, \theta, k, \boldsymbol{q}', t, f) \approx \boldsymbol{v}_{\text{STFT}}(t, f) \qquad (4)$$

We train our model using MSE loss between the generated and ground truth log-spectrograms $\boldsymbol{v}_{\text{STFT}}$:

$$\mathcal{L}_{\text{NAF}} = \|\Omega(\boldsymbol{q}, \theta, k, \boldsymbol{q}', t, f) - \boldsymbol{v}_{\text{STFT}}(t, f)\|^2 \qquad (5)$$

across spectrogram coordinates $t$ and $f$.

In our qualitative results we utilize Griffin-Lim reconstruction to recover $\boldsymbol{v}$ from $\boldsymbol{v}_{\text{STFT}}$ similar to prior work (Du et al., 2021). While such an approximate reconstruction ignores the underlying phase

of the impulse response, phase free models are typically used in spatial acoustic modeling (Pulkki, 2007; Raghuvanshi & Snyder, 2018; Singh et al., 2021). Such approximations are used because the phase locking cutoff in humans occurs at around $1\sim 2$kHz (Shinn-Cunningham et al., 2000; Joris & Verschooten, 2013), below which time delay provides important spatial cues.

We demonstrate in our qualitative results that NAFs with this approximation may still accurately model realistic spatial acoustic effects. Alternative methods for waveform recovery exist, including methods that can be learned from data (Oord et al., 2016; Kalchbrenner et al., 2018). These approaches are orthogonal to our work, and we leave this exploration to a future study.

### 3.3 Generalization through Local Geometric Conditioning

We are interested in parameterizing the underlying acoustic field, so that we may not only accurately represent impulse-response at emitter-listener pairs we see during training, but also at novel combinations of emitter and listener seen at test time. Such generalization may be problematic when directly parameterizing NAFs utilizing a MLP with inputs specified in Eqn (4), as the network may learn to directly overfit and entangle the relation between emitter and listener impulse-responses.

What generic information may we extract from a given impulse-response between an emitter and listener? In principle, extracting the full dense geometric information in a scene would enable us to robustly generalize to new emitter and listener locations. However, the amount of geometric information available in a particular impulse-response, especially for positions far away from either current emitter and listener is limited, since these positions has little impact on the underlying impulse-response. In contrast, the local geometry near either emitter and listener positions will have a strong influence in the impulse-response, as much of the anisotropic reflection comes from such geometry (Paasonen et al., 2017). Inspired by this observation, we aim to capture and utilize local geometric information, near either emitter or listener locations, as a means to predict impulse-responses across novel combinations.

To parameterize and represent these local geometric features, we learn a 2D grid of of spatial latents which we illustrate in Figure 2. When predicting an impulse-response at a given emitter and offset position, we query the learned grid features at both emitter and listener positions using bilinear sampling, and provide it as additional context into our NAF network $\Omega$. Such features provide rich information on the impulse-response, enabling NAF to generalize better to unseen combinations of both emitter and listener locations. In the rest of this work, we refer to the NAFs with local geometric features as $\Omega_{grid}$. We learn grid latent features jointly with the underlying parameters of NAF. Additional details can be found in Appendix B.

Such a design choice, however, still requires us to consider how to further combine local geometric information captured separately from either listeners or emitters. A naïve implementation would be to maintain separate feature grids for both listener and emitter positions. Such an approach fails to account for the fact that the local geometric information captured by emitter may also inform the local geometric information around a listener. Examining Eqn (1), we note that it is in fact symmetric with respect to exchanging either listener or emitter positions (Chaitanya et al., 2020), indicating that the impulse-response does not change when listener and emitter are swapped. Such a result means that we may in fact directly utilize the local geometric information captured near an emitter position interchangeably for either emitters and listeners. Thus, we propose our local geometric information as a single latent grid, which we show in our ablations outperform the naïve dual grid implementation.

## 4 Experiments

In this section, we demonstrate that our model can faithfully represent the acoustic impulse response at seen and unseen locations. Additional ablation studies verify the importance of utilizing local geometric features to enable test time generation fidelity. Next, we demonstrate that learning acoustic fields could facilitate improved visual representations when training images are sparse. Finally we show that the learned NAF can be used for sound source localization.

### 4.1 SETUP

For evaluating the learned acoustic fields, we use the Soundspaces dataset (Chen et al., 2020). This dataset consists of $R_i$ probe points for each scene, with each probe capable of representing an emitter or listener location for up to $O(R_i^2)$ emitter and listener pairs. The emitters are represented as omnidirectional, while the listener acts as a stereo receiver that can have one of four different orientations. The listeners and emitters are at fixed height. For each scene, we holdout $10\%$ of the RIRs randomly as a test set.

Our NAFs are trained on 6 representative scenes, selected such that 2 consist of multi-room layouts, 2 consist a single room with a non-rectangular walls, and 2 consist of a single room with rectangular walls as in Figure 4. Each scene is trained for 400 epochs, which takes around 3.5 hours for the largest scenes on four `Nvidia V100s`. In each batch, we sample 20 impulse responses, and randomly select $2,000$ frequency & time pairs within each spectrogram. An initial learning rate of $5 \times 10^{-4}$ is used for the network, while an initial learning rate of $1 \times 10^{-3}$ is used for the grid. We add a small amount of noise sampled from $\mathcal{N}(0, 0.02)$ to each coordinate during training to prevent degenerate solutions.

### 4.2 ARCHITECTURE DETAILS

The Soundspaces dataset lacks the full parameterization of an acoustic field described in Equation 4, so we train NAF with a restricted parameterization that is available in the dataset. This allows for two degrees of freedom along the $x - y$ plane for the listener locations $q \in \mathbb{R}^2$ and the emitter location $q' \in \mathbb{R}^2$. The listener can assume four possible orientations $\theta \in \{0, 90, 180, 270\}$, while the emitter is omnidirectional. In particular, we utilize a parameterization of $\Omega_{\text{grid}}$ which maps an input tuple $[x, y, x', y', f, t] \in \mathbb{R}^8 \times \{0, 90, 180, 270\} \times \{0, 1\}$ to a single scalar value that represents the intensity for a given time and phase in the STFT:

$$\Omega_{\text{grid}}(x, y, \theta, k, x', y', t, f) \Rightarrow \boldsymbol{v}_{\text{STFT}}(t, f) \tag{6}$$

To encode the rotation $\theta$, as there are only 4 possible discrete rotations in the dataset, we directly query into a learnable embedding matrix of shape $\mathbb{R}^{4 \times k}$, returning a $\mathbb{R}^{1 \times k}$ vector. Similarily, to encode the left and right ear, we similarly query into a learnable embedding matrix of shape $\mathbb{R}^{2 \times k}$, returning a $\mathbb{R}^{1 \times k}$ vector. The $f, t$ tuple representing the frequency and time respectively are scaled to $(-1, 1)$ and processed with sinusoidal encoding using 10 frequencies of sin and cos up to 200Hz.

To obtain local geometric features for either a emitter or listener in a scene, we assume that our scene is contained with a set of bounding pixels $\mathcal{P} = \{P_1...P_k\}$ which form a grid over the scene. For a given position tuple $(x, y)$, we then query the local features:

$$(x, y) \Rightarrow \mathcal{L}(x, y; \tilde{f}(p_1^*), \tilde{f}(p_2^*), \tilde{f}(p_3^*), \tilde{f}(p_4^*)) \tag{7}$$

Where $\mathcal{L}(\cdot)$ is the bilinear interpolation function, $(p_1^*...p_4^*)$ are the four vertices that bound $(x, y)$, and $\tilde{f}(\cdot)$ represents the features stored at a given vertex. These queried features are combined with the coordinates processed with sinusoidal encoding using 10 frequencies of sin and cos functions up to 100Hz. We process both the listener and emitter position tuples this way. We combine the grid based features with the sinusoidal embeddings and the discrete indexed embeddings as the input to our multilayer perceptron $f_\phi$. Please refer to Figure 2 for a visualization of our model, and Appendix B for further details. We compare using a shared local geometric feature with the emitter and listener, as well as using have the emitter and listener query their own individual grids.

### 4.3 EVALUATION ON NEURAL ACOUSTIC FIELDS

We first validate that we can capture environmental acoustics with high fidelity, at unseen emitter-listener positions.

**Baselines.** We compare our model against several strong interpolation baselines. We experiment with holding either the listener fixed and interpolating based on the emitter location, as well as holding the emitter fixed and interpolating based on the listener location. Both linear and nearest neighbor approaches are widely used (Savioja et al., 1999; Raghuvanshi et al., 2010; Pörschmann et al., 2020) in modeling of spatial audio. We also compare sharing and using individual local geometric features.

| Model | Large 1 | | Large 2 | | Medium 1 | | Medium 2 | | Small 1 | | Small 2 | | Mean | |
|---|---|---|---|---|---|---|---|---|---|---|---|---|---|---|
| | MSE↓ | T60↓ | MSE↓ | T60↓ | MSE↓ | T60↓ | MSE↓ | T60↓ | MSE↓ | T60↓ | MSE↓ | T60↓ | MSE↓ | T60↓ |
| Linear-Interp | 3.349 | 7.357 | 3.831 | 8.975 | 3.108 | 7.515 | 3.425 | 7.958 | 3.601 | 9.741 | 6.487 | 11.62 | 3.967 | 8.862 |
| Nearest-Interp | 3.208 | 6.738 | 3.568 | 7.897 | 2.813 | 4.526 | 2.811 | 4.902 | 2.691 | **4.198** | 2.625 | 3.963 | 2.952 | 5.371 |
| NAF (Dual) | 1.399 | 4.311 | 1.423 | **4.880** | 1.376 | 5.004 | 1.383 | 4.200 | 1.398 | 6.687 | **1.533** | 3.801 | 1.419 | 4.814 |
| NAF (Shared) | **1.395** | **4.160** | 1.421 | 5.247 | **1.340** | **3.801** | **1.321** | **3.886** | **1.397** | 6.752 | 1.534 | **3.770** | **1.401** | **4.602** |

Table 1: **Quantitative Results on Test Set Accuracy.** We report the MSE difference between generated and ground truth log spectrograms across methods, as well as the percentage (%) difference for the T60 reverberation time. The best method for each room is **bolded**. For the nearest and linear baselines, we perform interpolation in the time domain using samples from the training set.

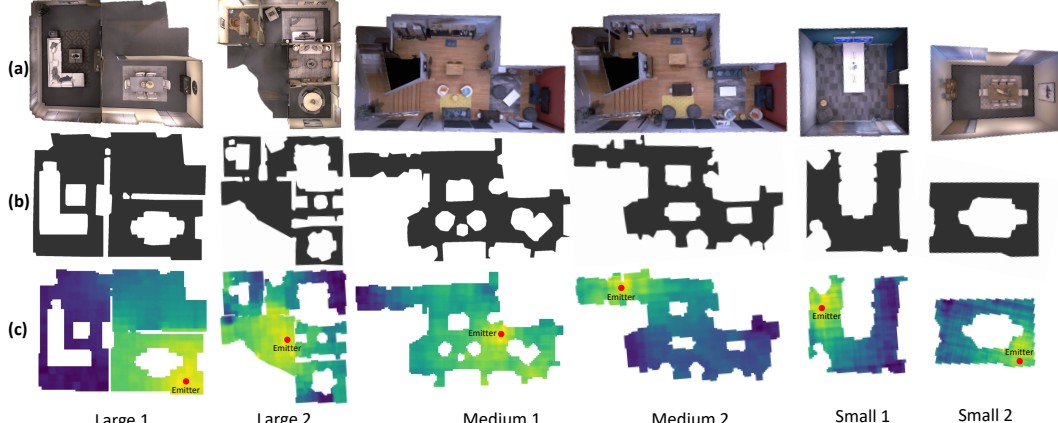

Figure 4: **Qualitative Visualization of Neural Acoustic Fields.** **(a)** The 3D structure of the room. **(b)** Walkable regions shown in grey. **(c)** Strength of predicted impulse response given a emitter location, lighter color indicates louder sound.

**Results.** We evaluate the results of our synthesis by measuring the mean squared error (MSE) between the generated and the ground truth log-spectrograms, as well as measuring the percentage error between the T60 reverberation time in the time domain. In this case, lower MSE and T60-error values indicates a better result. As shown in Table 1, our NAFs achieve significantly higher quality on the modeling of unseen impulse responses compared to strong interpolation baselines. A comparison of using shared and dual local geometric features indicates that despite having fewer learnable parameters, we achieve better performance by sharing the local geometric features. Examples of individual impulse responses generated by our model are shown in Figure 3. Figures 4 shows the different scenes and the loudness change predicted by our NAFs. Our model is capable of predicting smoothly varying acoustic fields that are affected by the physical surroundings.

**Generalization through Geometric Conditioning.** We next assess the impact of utilizing local geometric conditioning as a means to generalize to novel combinations of emitter-listener positions. On the "Large 1" room, in Figure 5 we evaluate test set MSE error when NAF is trained with a limited percentage of the training data either with or without local geometric conditioning. We find that such geometric conditioning enable substantially better test set reconstruction error, with the underlying performance gap increasing with less data.

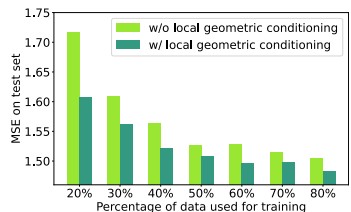

Figure 5: **Local Geometric Conditioning.** Comparison of NAF with and without local geometric conditioning trained with different amounts of data.

### 4.4 CROSS-MODAL LEARNING

When NAF is trained, as shown in Figure 6, a structured geometric representation emerges. This structured latent stands in sharp contrast with the noisy structure learned by NeRF models. In this experiment, we explore the effect of jointly learning acoustics and visual information when we are given sparse visual information. Recall that our NAF includes a local geometric feature grid $\mathcal{P}$ that covers the entire scene. For our cross-modal learning experiment, we jointly learn this feature grid with a NeRF network modified to accept both local features along with the traditional sinusoidal

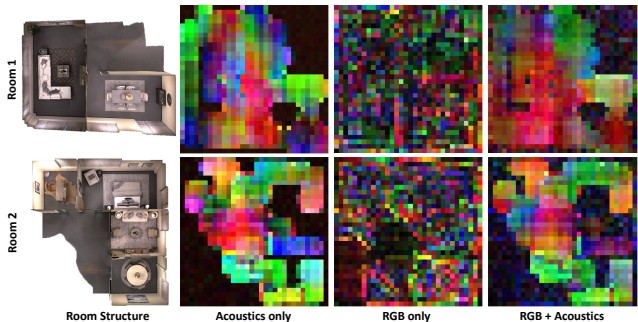

Figure 6: **Qualitative Grid Visualization.** Visualization of the learned grid features with principal component analysis. When trained using acoustic information, the grid learns highly structured information. When using NeRF, the grid is highly noisy. Structure emerges when we train both frameworks together.

embedding. In the acoustics branch, we query the grid using emitter and listener positions. In the NeRF branch, we use point samples along the ray projected on the grid plane to query the features. In both cases, the process is fully differentiable. We use a standard implementation of NeRF with a coarse and fine network. In both the cross-modal and RGB only experiments we augment the fine network with a learnable local feature grid. In the NeRF only setting, we minimize color $C$ reconstruction loss for a ray $r$ over a batch of rays $\mathcal{R}$: $\mathcal{L}_{\text{RGB}} = \sum_{r \in \mathcal{R}} ||\hat{C}(r) - C(r)||_2^2$. In contrast, in the NAF + NeRF experiment, we jointly minimize $\mathcal{L}_{\text{RGB}} + \mathcal{L}_{\text{NAF}}$, where $\mathcal{L}_{\text{NAF}}$ is defined in equation 5. We utilize 64 coarse samples and 128 fine samples for each ray, and sample 1024 rays per batch.

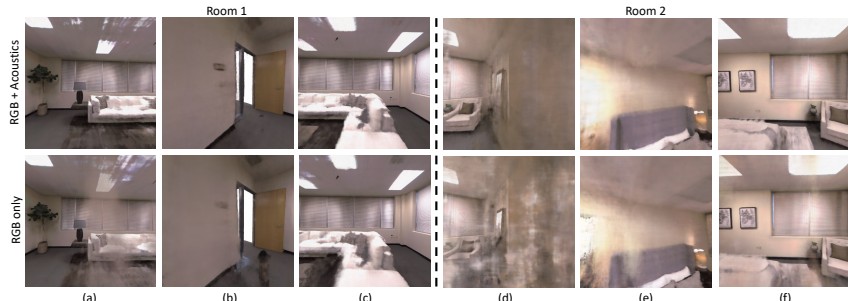

Figure 7: **Qualitative Visualization of Cross-Modal Image Generation.** Qualitative comparison between NeRF learned jointly with a NAF with RGB and acoustic supervision, and NeRF learned with only RGB supervision. We observe fewer floating artifacts when jointly training with audio. **(a)-(c)** Three views from "Large 1". **(d)-(f)** Three views from "Large 2".

| | Large Room 1 | | | | | | Large Room 2 | | | | | |
|---|---|---|---|---|---|---|---|---|---|---|---|---|
| | PSNR ↑ | | | MSE ↓ | | | PSNR ↑ | | | MSE ↓ | | |
| Training Images | 75 | 100 | 150 | 75 | 100 | 150 | 75 | 100 | 150 | 75 | 100 | 150 |
| NeRF | 25.41 | 27.36 | 29.85 | 6.618 | 3.506 | 1.740 | 25.70 | 27.74 | 29.34 | 6.921 | 3.905 | 2.185 |
| NeRF + NAF | **26.19** | **27.59** | **29.90** | **5.209** | **2.983** | **1.625** | **26.24** | **28.22** | **29.45** | **5.641** | **3.075** | **2.034** |

Table 2: **Quantitative Results on Cross-Modal Image Learning.** Quantitative results on joint training of NeRF and NAF jointly conditioned on a single local grid. We use very sparse training images in highly complex scenes. When evaluated on 50 test images, we observe that cross-modal learning helps improve PSNR when the visual training data is more sparse. MSE results are multiplied by $10^3$.

**Results.** We train on the two large rooms in our training set. For each room 75, 100, 150 images are used for training, while the same 50 images of novel views are used for all configurations during testing. In Table 2 we observe that training with acoustic information helps improve the PSNR and MSE of the visual output. This effect is more significant when the training images are very sparse, the NAF network helps less when there is sufficient visual information. Qualitative results are shown in Figure 7, we see there is a reduction of floaters in free space.

We visualize the local features learned in Figure 6. While the features learned using only RGB supervision lack coherent structure, the one we jointly learn with audio exhibits clear structure. To assess the structure in grid latents, we fit a linear layer to regress the room position of each latent. For

room 1 and 2 respectively, we achieve a $R^2$ of 0.109 and 0.085 when trained on RGB only, compared to a $R^2$ of 0.385 and 0.201 when trained with RGB + Acoustics supervision. This demonstrates the spatially informative structure captured by jointly learning NAF and NeRF.

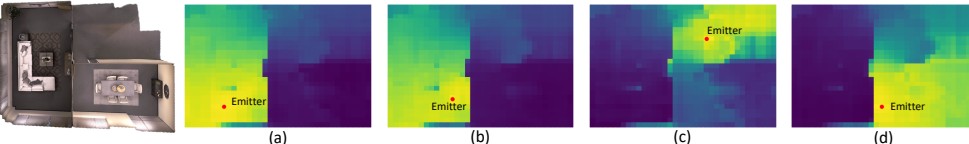

Figure 8: **Qualitative Visualization of Sound Localization.** Our NAFs can localize where a sound is coming from given listener samples. **(a)-(d)** Given unperturbed audio, the NAF can model the sound by placing the source anywhere in the scene. For each grid location as emitter, we evaluate how well the listener response matches the observation. The actual emitter location is shown in red. Lighter color indicates that if an emitter was placed here, there is lower error compared to observations, which matches well with the real emitter location.

| Method | Chords | Bells | Flute | Piano | Sweep |
|---|---|---|---|---|---|
| ResNet-18 | 0.658 | 0.476 | 0.916 | 0.983 | 0.188 |
| Max mag. | 0.126 | 0.393 | 0.919 | 0.225 | 0.531 |
| NAFs | **0.004** | **0.025** | **0.038** | **0.004** | **0.015** |

Table 3: **Quantitative Results on Sound Localization.** Quantitative results on sound localization distance in normalized room coordinates. NAFs can accurately estimate location of the emitter.

## 4.5 SOUND SOURCE LOCALIZATION

Given a high fidelity representation of the acoustic field, it is possible to identify the location of a sound source in a scene, regardless of where the listener is located? In this task, we demonstrate that a trained NAF can be used for localizing an emitter based on sparse audio samples from the scene. We explore 5 sounds in total, 4 of which are musical instruments and 1 is a sine sweep (Sweep).

We assume that exists a set of audio recordings $V_i \in \mathcal{V}$ from an object, recorded from 10 different locations and orientations that are known. We further assume that we are given the original unperturbed waveform $V_{\text{orig}}$ the object emits, with the underlying task being to locate the position of the emitter $(x', y')$ using these sound samples and the unperturbed sound.

We hold the listener position and orientations fixed and vary the emitter position to probe for the expected sound at each location. For each location, we render a log-spectrogram using our trained NAF and compare this against the actual log-spectrogram recorded response $V_i$:

$$|| \log(\text{stft}(V_{\text{orig}} \circledast p_i(t; \boldsymbol{x}'))) - \log(\text{stft}(V_i))||^2 \tag{8}$$

We perform the search for the emitter location by discretizing the scene in $0.3m$ segments.

**Baselines.** We compare against two baselines, a ResNet-18 network that takes as input both the spectrogram of the transformed and original sounds, and is trained to regress the object location relative to the listener frame. Given 10 individuals, we take an average of the predicted emitter location. We also compare against a modified maximum magnitude baseline, where given a set of recording locations, we simply select the position where the sound is loudest.

**Results.** Our NAFs can achieve very accurate localization as seen in Table 3. We also visualize the error map with the ground truth emtiter location labeled in Figure 8. It can be observed that the real location of the emitter matches well with the location of the lowest prediction error.

## 5 CONCLUSION

In summary, this paper introduces Neural Acoustic Fields (NAFs), a continuous, differentiable acoustic representation which can faithfully represent the reverberation of different audio sources in a scene, as well as the scene's overall acoustic environment. By conditioning NAFs locally on the underlying scene geometry, we demonstrate that our approach enables the accurate prediction of environmental reverberations even at unseen locations in the scene. Furthermore, we demonstrate that the acoustic representations learned through NAFs are powerful, and may be utilized to facilitate audio-visual cross-modal learning, as well as to localize the underlying source of arbitrary sounds.

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

# Appendix

## A ADDITIONAL VISUALIZATION OF ROOMS

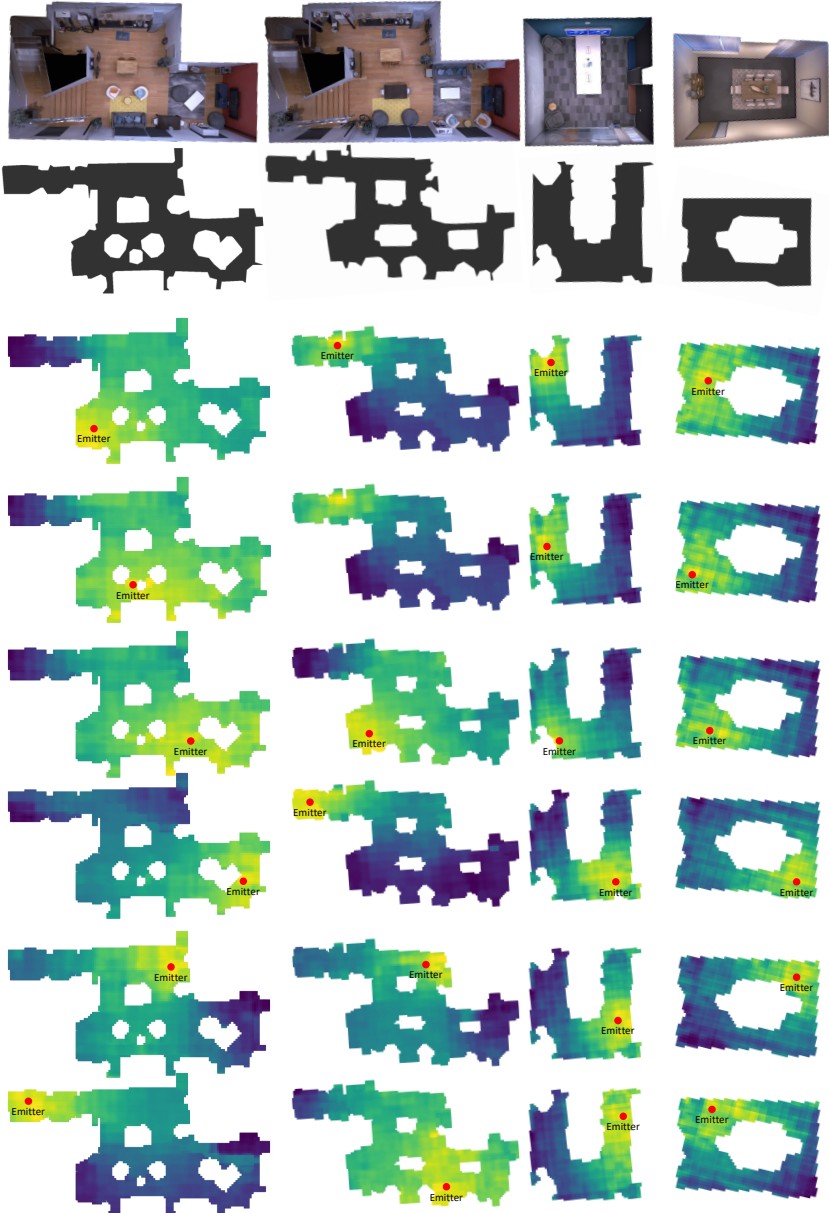

Figure A1: **Additional Qualitative Predictions of NAF.** Qualitative visualization of the loudness map as predicted by NAF across four different rooms.

We show additional NAF predictions of loudness as we move an emitter inside different rooms in Figure A1. For each room, note how the sound is affected by the geometry. In wide open spaces the sound is highly dispersed. While in thin structures the sound test to concentrate locally. As we move farther from the source, the loudness of the sound decreases.

## B    ARCHITECTURE AND TRAINING DETAILS

We visualize the two alternative models that we experiment with, in Figure A2 is a network that uses different local feature grids for the emitter and receiver. The network uses the emitter and listener to positions to sample from the two different grids.

In Figure A3 we show a model that does not utilize any kind of local geometry conditioning. The listener, emitter, phase, and time input are transformed using sinusoidal embedding, while the orientation and left/right are retrieved. All transformed inputs are directly fed to the network.

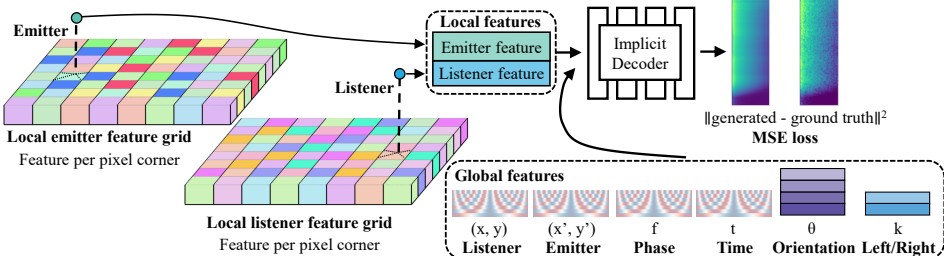

Figure A2: Architecture of the model that uses emitter and listener specific local geometry conditioning.

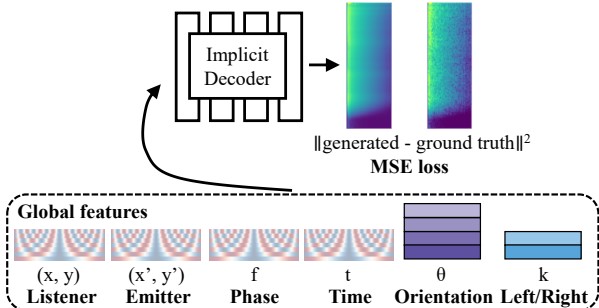

Figure A3: Architecture of the model that uses no local geometry conditioning.

Each network consists of 8 fully connected layers in a feedforward fashion, as well as a skip connection consisting of two fully connected layers. The skip connection takes the input and adds its output to that for the fourth intermediate layer. We utilize an intermediate feature size of 512, and softplus as the activation function. The grid is of size $\mathbb{R}^{32 \times 32 \times 64}$, where 32 is the size of the spatial dimensions, while 64 is the channel count. We initialize each element of the grid i.i.d. from $\mathcal{N}(0, \frac{1}{\sqrt{64}})$. For the network excluding the grid, we utilize an initial learning rate of $5e - 4$. For the grid, we utilize an initial learning rate of $1e - 3$. The *Adam* optimizer is used when training our network. We utilize a orientation embedding of shape $\mathcal{R}^{7 \times 4 \times 512}$ where 7 is the number of intermediate outputs, 4 is the number of orientations, and 512 is the feature dimension. For the left-right embedding, we use a shape of $\mathcal{R}^{7 \times 2 \times 512}$. We perform additive conditioning by adding a $\mathcal{R}^{512}$ vector to each intermediate output for both the orientation and the left/right.

For each scene, to generate a log-spectrogram for each impulse response, we compute the mean and standard deviation $\mu_{\text{freq}}, \sigma_{\text{freq}}$ for each frequency in the log-spectrogram, and normalize the data prior to training:

$$v_{\text{freq}} = \frac{v_{\text{freq}} - \mu_{\text{freq}}}{3.0 \times \sigma_{\text{freq}}}$$

For the sinusoidal embedding, we utilize both $\cos$ and $\sin$ with 10 frequencies each for encoding position, phase, and time. For encoding position we utilize a max frequency of 100Hz, while for encoding time and frequency we utilize a max frequency of 200Hz.

Since we do not know beforehand the time duration of an impulse response at an unseen location, we compute the maximum impulse length for each scene and use this length to zero pad the training

impulse responses. Because the padded regions do not contain useful information, we want the network to focus modeling efforts on the early regions of the impulse response. We achieve this by stochastically padding the impulse response to maximum impulse length with $0.1$ probability. Because the implicit function is trained on individual $(t, f)$ coordinates within a given $v_{\text{STFT}}$, training samples do not need to be of the same length. During test time, we perform inference up to the maximum duration of scene impulse response.

## C  DATASET VISUALIZATION

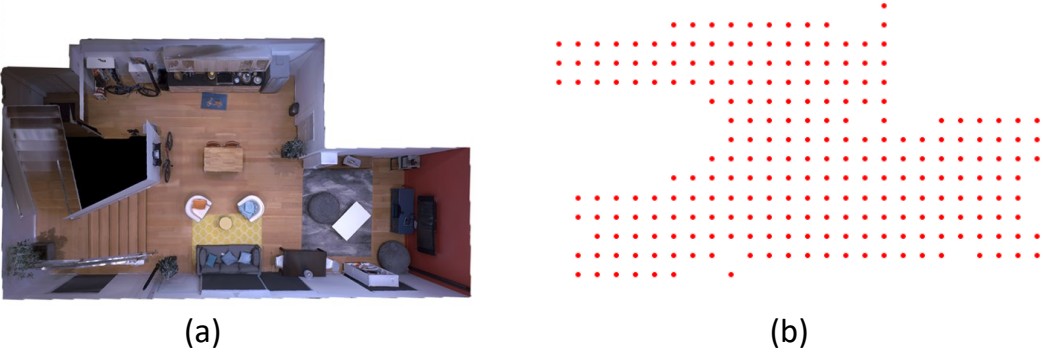

(a)                                                                                         (b)

Figure A4: A room the emitter-listener probes. **(a)** The 3D structure of a room. **b** The probes marking the location of emitters/listeners.

In Figure A4, we visualize both the room and underlying set of probe positions in the training data. Due to occlusion and the geometry, even slightly moving the emitter or listener position can result in different results. As we demonstrated in Table 7, both nearest neighbor and linear interpolation perform poorly compared to our learned solution. In contrast, recovered acoustic fields from NAF trained on these probe positions is substantially denser (Figure A1).

## D  STORAGE COMPARISON

| Method | Storage (MB) |
|---|---|
| Linear-Interp | 5970 |
| Nearest-Interp | 5970 |
| NAFs (Dual Local Feat) | 27.07 |
| NAFs (Shared Local Feat) | 25.37 |

Table A1: **Approximate on disk storage cost of different methods.** We average the amount of data required for different methods of inference for the six scenes. Note that the linear and nearest interpolation methods require the entire training set, while the NAF based methods use constant storage.

We compare the averaged on disk storage cost of the different methods for inferring the spatial audio using a precomputed training set in Table A1. Both linear and nearest interpolation methods require access to the entire training set, while our NAF based approaches compactly encode the acoustic scene into a fixed size.

## E  VISUALIZATION OF INTERPOLATED METHODS

In this section, we visualize the loudness when an interpolation method is used. We compare nearest neighbor interpolation with linear interpolation applied to the training set, both in the time domain. The nearest neighbor method presents sudden step changes in loudness, while the linear interpolation fails when the queried location falls outside of the convex hull of the training set. Also note the sound leakage in **(c)-(d)** in the bottom wall when linear interpolation is used.

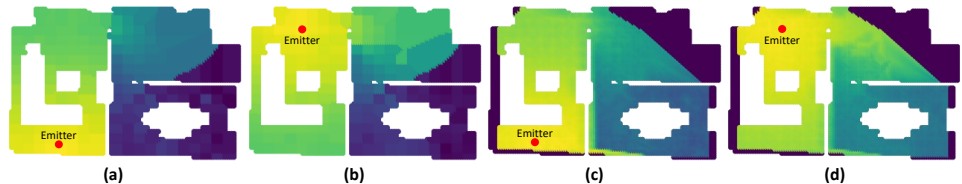

Figure A5: **Visualization using interpolated sound.** We show the loudness at different locations using **(a)-(b)** nearest-neighbor interpolation and **(c)-(d)** linear interpolation.

## F  ALTERNATIVE REPRESENTATIONS

| Representation | MSE | T60 |
|---|---|---|
| Magnitude only | 1.395 | 4.160 |
| Magnitude + phase | 1.439 | 5.254 |
| Time domain | 39.87 | 61.82 |

Table A2: **Learning different representations** We compare learning magnitude only, jointly learning magnitude and phase, as well as directly learning in the time domain.

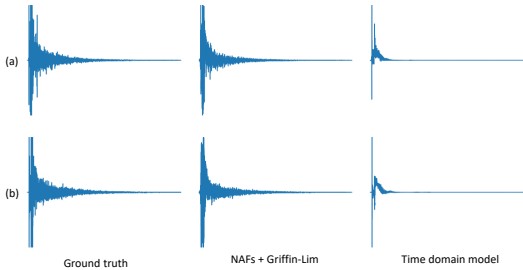

Ground truth          NAFs + Griffin-Lim          Time domain model

Figure A6: **Qualitative Visualization of impulse waveforms at unseen emitter/listener locations.** From left to right, we show the ground truth waveform, the Griffin-Lim recovered waveform, and the waveform learned by a network in the time domain for two locations **(a)-(b)** not seen in training.

Our current method follows prior work in learning in the log-magnitude STFT domain. In this section, we investigate two possible alternatives: learning phase + log-magnitude, and directly learning in the time domain. The MSE and T60 error percentage is presented in Table A2. We observe that jointly modeling phase + log-magnitude degrades the performance slightly compared to modeling just the log-magnitude, while modeling in the time domain performs poorly. We visualize the waveform at two test set listener/emitter locations in Figure A6.

## G  $L_2$ REGULARIZED GRID IN NERF

|  | Large 1 | | Large 2 | |
|---|---|---|---|---|
|  | PSNR ↑ | MSE ↓ | PSNR ↑ | MSE ↓ |
| NeRF + grid + $L_2$ | 22.69 | 6.956 | 24.86 | 7.128 |
| NeRF + grid | **25.41** | **6.618** | **25.70** | **6.921** |

Table A3: **Regularizing the grid.** In this experiment, we compare learning NeRF with a grid without regularization, and with $L_2$ regularization.

In Table A3 we compare NeRF that utilizes a grid and trained using image reconstruction loss, against a variant where a $L_2$ penalty with weight $1e-5$ to ensure a smooth latent space is added to the image reconstruction loss. There are 75 images used in the training set. We observe degraded performance when we apply this penalty.

# H   Interpolation of left/right latent

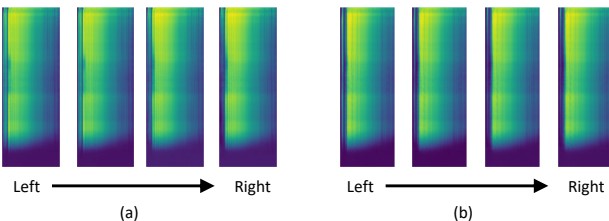

Figure A7: **We interpolate linearly between left/right latents.** Here we show two examples **(a)-(b)** where we take linear steps between the latent representing the left ear, and the latent representing the right ear. Pay attention to the onset (left) of each spectrogram.

As shown in Figure A7, are the output for two locations when we take linear steps between the latent representing the left ear, and the right ear. We observe that the output smoothly changes as we move in the latent space.

