# OpenReview forum: "Learning Neural Acoustic Fields"
_ICLR.cc/2022/Conference — ICLR 2022 Submitted_

### Official Review · Reviewer_U1r6 · 2021-10-24

**Correctness:** 3
**Technical Novelty And Significance:** 2
**Empirical Novelty And Significance:** 2
**Recommendation:** 3
**Confidence:** 4

**Main Review:**

The manuscript rightfully points to the importance of being able to roughly deduce the scene geometry from monaural recordings. To address this challenge, the authors propose a neural network architecture that not only takes into account the speaker and listener positions in a room but also some geometrical information as well. Overall, while I do think this is an important and challenging problem, I think the manuscript falls short of what it promises. I also found the manuscript unclear about important details at times. Overall, it should do better in terms of presenting convincing evidence to show that the neural network does more than learning to interpolate a smooth field (since log magnitudes are the only outputs of the neural network).

Here are some more specific comments :

1. Introduction, third paragraph : I agree that being able to infer scene properties given monaural recordings is something humans do, but I think the proposed method is a rather modest step in this direction. After all, for each individual room, you need to have many prerecorded room impulse responses to train the neural network. When you enter a totally new room (with no previous RIRs available), it's not clear to me how one could use the proposed neural network to infer the shape of the room. Could you elaborate?

2. Page 2, "... we propose to condition NAFs on the local geometric information present at both the listener and emitter locations..." : I understand you provide a grid with the emitter, listener locations, as in Fig 2, but how do you form the grid exactly -- you write about this verbally, but can you be more specific about the details?

3. eqn (3) : It's interesting that you encode binary information (which ear) as a real. I'm curious, what happens when you input $k = 0.5  (k_{\text{left}} + k_{\text{right}})$?

4. In eqns (3), (4), please indicate the dimensions of the components (i.e., $x\in\mathbb{R}^3$, $\theta \in \mathbb{R}^2$ etc.)

5. It's a bit confusing to say that $\Phi$ in eqn (4) is your NAF -- it's the version without geometric features, isn't it? I'd suggest declaring another function, and clearly stating the input dimensions of the various arguments, so as to separate it from $\Phi$.

6. For $\Phi$ in eqn (4), you try to predict the logarithm of the magnitude -- how do you recover the phase? Isn't that an important part of the room impulse response?

7. Section 4.3, "...pad the training impulse response with 0.1 chance." : What does "padding with a chance" mean? I thought you'd zero pad short impulse responses so that the length of every impulse response is the same.

8. Section 4.3 : How do you select the test set? Do you randomly sample the available RIRs? Or do you hold out a specific region entirely? The latter would be more interesting in terms of evaluating the generalization capability, touched upon in the third paragraph of the Introduction (see my comment-1).

9. Eqn (8) : Isn't the idea to obtain $p_i$ from the NAF? How do you deal with the lost phase of the impulse response (see comment-6)?

10. Section 4.5, baselines : I understand you minimize eqn (8) over $x$, right? If so, for the nearest neighbor method, are you doing a search over the available room impulse responses? I don't understand why you'd choose the loudest. Please clarify.

**Summary Of The Paper:**

The manuscript considers the problem of encoding room impulse responses in a room. Specifically, the authors propose a network that takes in some geometrical cues from the room, along with speaker, listener positions, and train a neural network using pre-recorded room impulse responses. They argue that the neural network learns to make use of the geometrical cues, and can generalize beyond the dataset.

**Summary Of The Review:**

Overall, the manuscript considers a very interesting problem, but I am not convinced that the proposed method would be sufficient to provide a solution as implied in the Introduction. Specifically, given a large database of room impulse responses, the neural network appears to learn how to store a smooth field based on those values (since log magnitude is the target), but it's not really clear how the geometrical side information could be useful for generalizing beyond a room. I also found the paper to be a light on some of the details, which I think are critical.

---

> ### Author Response · Authors · 2021-11-14
> **Response to Reviewer U1r6 (Part 1/2)**
>
> We are encouraged by your assessment that this is an important and interesting question, and appreciate the detailed feedback. We will incorporate all suggestions into our paper. Below are our responses to specific comments.
>
> > **Q1) Data and generalization.**
>
> * **[Scope of our work]** Our work has the same scope as the corresponding visual models (NeRF, SRN, etc.), which fits a network to an *individual* scene, and seeks to achieve generalization to novel views (in our case novel listener/emitter positions) from sparse training data in the *same* scene [1, 2]. Modeling of completely unobserved locations using implicit networks is an open question in both vision and acoustics.
> * **[Generalizing in spatial audio]** In vision, one can leverage the multiview consistent nature of the scene to learn a dense representation using photometric loss. However we cannot make the same multiview consistent assumption in learning acoustic scenes. Instead, we leverage the fact that anisotropic reflections are strongly affected by local geometry [3]. We propose instead to generalize by explicitly learning the local geometric attributes.
> * **[Contribution of our work]** Our work represents the first step towards representing spatial acoustics using a neural network. We show that our system is significantly better than nearest neighbor and linear interpolation baselines. We look forward to exploring generalization across scenes to a future work.
>
> We qualitatively demonstrate our ability to generalize to continuous locations not in the training set on our website, which we reproduce below:
> https://sites.google.com/view/nafs-iclr-2022
>
> > **Q2) Parameterization of the grid.**
>
> Thank you for the question. We provided details about the grid in section B of the appendix. We provide additional details here.
>
> We initialize a grid with 32x32 resolution and 64 feature maps. Each element of the grid is initialized independently from a gaussian with 0 mean and 1/sqrt(64) variance. For a coordinate from the scene, we normalize the coordinate to between [-1, 1] using the axis aligned bounding box of the scene. The features are queried using these normalized coordinates with bilinear sampling.
>
> We can clarify this in the text by including these details.
>
> > **Q3) Blending left/right ear latents.**
>
> The idea is very interesting, and we will provide qualitative results in an upcoming revision of our paper. It should be noted that our system is independent of the choice of panning method. For our empirical results we utilize linear panning, but HRTF based panning can also be utilized.
>
> > **Q4) Dimension of the variables.**
>
> We agree that additional information could be provided for each variable we use. We will clarify the dimensions in the revision.
>
> > **Q5) Notation of the NAFs.**
>
> Thank you for pointing this out. We will modify our notation to clarify which symbol represents the network with the grid.
>
> > **Q6) STFT and phase.**
>
> In practice, many learned and traditional models for spatial audio modeling do not seek to represent the phase. These models use either random phase with learned log-magnitude STFT, random phase or minimum phase filters respectively [4,5,3].
>
> Our model outputs the magnitude. To recover the phase from the magnitude for our qualitative results, we utilize the redundancy present in the STFT representation.
>
> 1. Given the magnitude only STFT representation, we apply the widely used iterative Griffin-Lim algorithm to reconstruct the phase [6]. After reconstructing the phase, we can perform inverse STFT to derive the time domain (wave) representation of the impulse response.
> 2. After computing the time domain impulse response, we convolve with an audio sample to render the final result.
>
> This approach of modeling magnitude only while using Griffin-Lim to recover phase is also used in other recent papers on sound generation [7,8,9].
>
> We will include these details in an upcoming revision of our paper.

---

> > ### Author Response · Authors · 2021-11-14
> > **Response to Reviewer U1r6 (Part 2/2)**
> >
> > > **Q7) Temporal zero padding of the data.**
> >
> > We indeed pad the data, as not all impulse responses within a single scene are of the same length. A highly reverberant room will generally have longer impulse responses. Since the training data is very sparse, the ground truth length of the impulse is not defined for most locations in a scene.
> >
> > To avoid missing information in an impulse, we model an impulse response up to the maximum length of the impulse in a scene. However the vast majority of impulses are significantly shorter, and do not provide useful information in the padded portion. Each time a given impulse is presented to the network for training, we pad it with 10% probability. This allows the network to focus on modeling the energetic early stages of the impulse. Because we learn an implicit function, not all data samples have to be the same length during training.
> >
> > > **Q8) Selection of the test set.**
> >
> > We randomly select the test set. We do this to allow our network to uniformly observe all locations in a scene. Our goal is to generalize to continuous locations in a scene given only sparse training samples.
> >
> > In Figure 5, we observe that by learning a grid feature that is shared between the emitter and listener, we perform consistently better than a MLP even in the very sparse settings. It should be noted that corresponding implicit visual approaches (NeRF, SRN, etc.) are also fit to a single scene, and requires parts of a scene to be observed to allow for inference.
> >
> > > **Q9) Phase for spatial acoustics.**
> >
> > Please see our response to Q6.
> >
> > > **Q10) Clarification of equation 8.**
> >
> > This section indeed needs to be better described. Equation (8) is utilized when our NAF is present. The modified nearest neighbor baseline is using max magnitude and is not learned. Essentially, we select the listener which has the highest average sound pressure level that is closest to the location of the emitter. We will clarify the setup in an upcoming revision of the paper.
> >
> >
> > Thank you for the suggestions. We will include these details and change our notation to improve the clarity of our paper.
> >
> > We genuinely appreciate your advice and suggestions, and will include your feedback in a revision of our paper.
> >
> > [1] Mildenhall, Ben, et al. "Nerf: Representing scenes as neural radiance fields for view synthesis." ECCV (2020).
> >
> > [2] Sitzmann, Vincent, Michael Zollhöfer, and Gordon Wetzstein. "Scene representation networks: Continuous 3d-structure-aware neural scene representations." NeurIPS (2019).
> >
> > [3] Raghuvanshi, Nikunj, and John Snyder. "Parametric directional coding for precomputed sound propagation." SIGGRAPH (2018).
> >
> > [4] Singh, Nikhil, et al. "Image2Reverb: Cross-Modal Reverb Impulse Response Synthesis." ICCV (2021).
> >
> > [5] Pulkki, Ville. "Applications of directional audio coding in audio." ICA 2007.
> >
> > [6] Griffin, Daniel, and Jae Lim. "Signal estimation from modified short-time Fourier transform." IEEE ASSP (1984).
> >
> > [7] Du, Yilun, et al. "Learning Signal-Agnostic Implicit Manifolds." NeurIPS (2021).
> >
> > [8] Wang, Yuxuan, et al. "Tacotron: Towards end-to-end speech synthesis." Interspeech (2017).
> >
> > [9] Ren, Yi, et al. "Almost unsupervised text to speech and automatic speech recognition." ICML (2019).

---

> > > ### Comment · Reviewer_U1r6 · 2021-11-27
> > > **Response to authors**
> > >
> > > Thank you very much for the detailed responses.
> > > This clarified some of my concerns. However, as another reviewer pointed out in the Responses phase, Griffin-Lim may fall short of recovering an RIR in this scenario. I have used Griffin-Lim for reconstructing audio from STFT magnitude, and I'm aware it produces realistic sounding audio, when the magnitude is known exactly. However, as the other reviewer pointed out, it's not really a reconstruction mechanism with recovery guarantees. If you recover a slightly incorrect RIR, and try to deconvolve using that, I'd expect that to lead to serious artifacts in the dereverbed signal.
> > >
> > > I also liked the demo you pointed us to, but it would be helpful to place side by side, your results against methods using nearest neighbor or other interpolation techniques. In particular, for linear interpolation, I guess you're interpolating the magnitudes, right (and not the complex values)? If we're relying on Griffin-Lim, and doing only sound synthesis, we might as well follow the same approach for interpolation based methods.

---

> > > > ### Author Response · Authors · 2021-11-28
> > > > **Response to Reviewer U1r6**
> > > >
> > > > Thank you very much for the questions and additional feedback!
> > > >
> > > > > **How is linear interpolation computed?**
> > > >
> > > > As we clarify in the caption for Table 1., we actually adopt a stronger baseline through interpolation in the time/waveform domain. Prior work mentions interpolation in both the time and frequency domain as valid approaches [1].  We utilized linear interpolation in the log-magnitude STFT domain initially. However, after performing the interpolation in time domain, we observe lower MSE error and T60 error compared to interpolation in the log-magnitude STFT and utilizing Griffin-Lim for phase recovery for the T60 metric. For this reason, we believe that time domain interpolation is the stronger baseline.
> > > >
> > > >
> > > > > **Phase and spatial audio.**
> > > >
> > > > For the purposes of gaming/VR tasks, past work for spatial audio representations generally also do not model phase. In particular, prior work for spatial audio representation: Pulkki's DirAC [2], Raghuvanshi's parametric coding [3], Drori's image2reverb [4] do not encode phase and construct either random phase filters, minimum-phase filters or sample a random phase for log-magnitude STFT reconstruction respectively.
> > > >
> > > > > **Speech dereverberation.**
> > > >
> > > > Like other work, we model the spatial sound by convolving a clean source with the impulse generated by our system. We do not explore dereverberation via deconvolution in this work. But we agree that direct deconvolution using our magnitude only representation with inferred phase would not yield high quality dereverberation. A possible approach to produce perceptually reasonable de-reverberant speech/audio could be learning a network to produce dry speech/audio conditioned on the reverberant audio STFT and the NAF predicted magnitude STFT impulse response [5], instead of performing blind dereverberation. In this case, we can reconstruct the time-domain signal from this clean magnitude spectrogram estimate using Griffin-Lim or a neural vocoder (wavenet). Our work takes the first step on modeling magnitude of impulse responses, and we believe this will inspire many follow-up works on this exciting direction.
> > > >
> > > > > **Additional qualitative results.**
> > > >
> > > > Thank you for this great suggestion. We are currently working on additional qualitative results, this will take one or two days. We will provide an update once the results are posted.
> > > >
> > > >
> > > > Please let us know if you have any additional questions!
> > > >
> > > >
> > > > [1] Raghuvanshi, Nikunj, et al. "Precomputed wave simulation for real-time sound propagation of dynamic sources in complex scenes." SIGGRAPH (2010).
> > > >
> > > > [2] Pulkki, Ville. "Directional audio coding in spatial sound reproduction and stereo upmixing." Audio Engineering Society Conference (2006).
> > > >
> > > > [3] Raghuvanshi, Nikunj, and John Snyder. "Parametric wave field coding for precomputed sound propagation." SIGGRAPH (2014).
> > > >
> > > > [4] Singh, Nikhil, et al. "Image2Reverb: Cross-Modal Reverb Impulse Response Synthesis." ICCV (2021).
> > > >
> > > > [5] Han, Kun, et al. "Learning spectral mapping for speech dereverberation and denoising." TASLP (2015).

---

> > > > ### Author Response · Authors · 2021-11-29
> > > > **Website updated with direct comparison**
> > > >
> > > > Thank you again for the suggestion of a direct comparison.
> > > > We have updated our website with new qualitative results. The videos are generated using the same trajectory, with the interpolation baselines done on the training set also used for the NAF. The original training dataset used for interpolation is roughly 10 GB in size, while our NAF is significantly more compact at less than 30 MB.
> > > >
> > > > [https://sites.google.com/view/nafs-iclr-2022/home](https://sites.google.com/view/nafs-iclr-2022/home)
> > > >
> > > > * For the nearest neighbor approach, note the sudden loudness change at 0:28. When the underlying data is sparse, this approach can jump suddenly from one impulse response to another, leading to jarring changes in the spatial audio.
> > > > * For the linear interpolation approach, note how the energy on the opposite side of the wall incorrectly influences the sound hear on the other side, this causes the audio to sound very close even when we are separated by a wall. Also note that as we exit the convex hull defined by the training data, we cannot correctly infer the spatial audio.
> > > >
> > > > In comparison, our approach can continuously infer a plausible spatial audio at various locations.

---

> ### Author Response · Authors · 2021-11-26
> **Sincerely Look Forward to Your Feedback!**
>
> Dear Reviewer U1r6,
>
> Thank you again for your suggestions. As the deadline for the discussion is coming up, we would be happy to address any remaining questions.
>
> In our previous response, we have carefully studied your suggestions and made updates to our revision, which we summarize below:
> * We clarify the scope and goal of our work in the introduction.
> * We provide additional technical details on the dimension and initialization of the grid.
> * We added an experiment to the Appendix on blending the left/right latent.
> * We now detail the dimension of each variable, and modify the notation to make the meaning of each variable more obvious.
> * We now clarify which version of our NAF has a grid in our paper.
> * We provide additional details on how the waveform can be recovered without learning the phase.
> * We added more details on how we perform temporal padding of the training data.
> * We added additional details on how we select the test set.
> * We have clarified our use of Eqn 8. and modified the language to clarify how we perform localization.
>
> We would like to know if you have any additional comments or suggestions. We’d be very happy to do address any remaining questions that we can in the time remaining!
>
> Thank you for your suggestions!
>
> Best,
>
> Authors

---

### Official Review · Reviewer_HhWA · 2021-10-31

**Correctness:** 2
**Technical Novelty And Significance:** 2
**Empirical Novelty And Significance:** 2
**Recommendation:** 5
**Confidence:** 4

**Main Review:**

Strengths:

- The idea of using a neural network to represent the acoustic fields of how sound propagates in space is new and interesting.

- The paper is well motivated and it makes a lot of sense to represent neural acoustic fields or sound propagation as a neural network, which is differentiable.

- Leveraging what is learned in the neural acoustic fields neural implicit representation, the paper demonstrates several tasks that are pretty interesting, including cross-modal generation and sound source localization.

Weaknesses:

- First of all, although this paper is well-motivated and set the stage to introduce Neural Acoustic Fields, which is an implicit neural representation that captures how sound propagates in a physical scene, actually the network only *remembers a dataset*. The background of the room impuse response and environmental reverberation is nice, but the dataset of room impulse responses is captured/introduced in another paper (Chen et al. 2020), which is not the contribution of this paper. The neural network that this paper introduces is basically a NeRF network and the paper trains such a network to remember the dataset. Fm the introduction of the paper, it sounds like the paper is going to introduce a differentiable neural network design that can model ray tracing and sound propagation in a fully differentiable way, but actually it's not.

- There are multiple places the paper overclaims its contributions: For example, in the end of Audio Field Coding in related work, it is claimed that "the work allows a listener to move and experience sounds that come from anywhere in a scene and can represent the acoustic field continuously at high fidelity by directly learning from data". This is overclaimed, because this is not enabled by the design of the proposed neural network. It is enabled by the dataset, which uses simulation to model room impulse response. The paper only uses a network to remember this dataset.

- Related to the discussions above, the paper didn't discuss at all why it is necessary to use such an implicit representation. Why not just directly use the dataset instead?

- Providing grid features near both emitter and listener positions as additional context seems like a very strong assumption. It provides much extra info to the neural network.

- For equation 6, it is mentioned v is a single scale value taht represents the intensity. But isn't *v* should be the room impulse response? This is very confusing.

- For figure 4, it would be great to show sound ground truth from the dataset for comparison. I assume these maps can also be directly generated using the dataset.

- The writing of the paper can also be further improved. There are multiple places in the paper that are not clear or need more details. Here are some detailed questions:"

1) More descriptions are need on the local feature grid part. Currently the descriptions are very vague. It is not clear what exactly is the local features and what is used to represent them.

2) For Equation 3, it is unclear what theta and k is. Although it becomes clearer later when introducing the dataset, but the dataset is modeling a grid in 2d. So x and x' only have 2 dimensions instead. This should be clarified and discussed.

3) The RGB only baseline in Figure 6 is unclear and there is no description of the baseline.

4) How joint training is done for cross-modal image generation is also unclear. What the NeRF baseline is trying to encode?

**Summary Of The Paper:**

This paper introduces the concept of Neural Acoustic Fields, which is an implicit representation to capture how sound propagates in a pysical scene. Using a NeRF-like network architecture, the model is able to predict room impulse response given the listener and source locations as input. The idea is new and interesing, but there are multiple overclaims in the paper. Some interesting tasks are demonstrated like sound source location, cross-modal generation, etc.

**Summary Of The Review:**

In general, this paper studies an interesting problem and presents some preliminary attempts. But overall, the proposed method seems just another way to represent an existing dataset from another work. The paper overclaims its contribution and there are multiple places in the paper that needs more clarifications or analysis or writing improvement. Therefore, I recommend reject of the paper at this stage but happy to discuss more.

---

> ### Author Response · Authors · 2021-11-14
> **Response to Reviewer HhWA (Part 1/2)**
>
> We appreciate reviewer's assessment that the study is new and interesting. Thank you for the suggestions. Below are our responses to specific concerns.
>
> > **Q1) Comparison to NeRF and ray tracing.**
>
>
> We agree it is important to highlight the difference between our approach and NeRF. Our work utilizes **fundamentally different assumptions** compared to visual domain modeling. Our work is the first to attempt the modeling of spatial impulse responses with a neural network.
>
> * **[Scope of our work]** To clarify, like corresponding visual models (NeRF, SRN) our network is fit to a *specific* scene, and seeks to achieve generalization at unseen emitter/listener locations (or novel views if in the NeRF framework) from sparse training samples in the *same* scene [1,2]. The learning of implicit representations that can generalize to unseen scenes is an open question in both vision and acoustics, and we look forward to exploring this in future work.
> * **[Generalizing to unseen locations]** NeRF leverages the multiview consistent nature of the visual world to learn a dense scene from sparse samples using photometric loss. However, multiview consistency cannot be assumed in spatial acoustic modeling. Because the anisotropic reflections are strongly affected by local geometry [3], we instead propose to learn the necessary local geometric attributes of a scene. We demonstrate in Figure 5. that by learning this geometric feature, we can generalize to unobserved locations significantly better than a simple MLP.
> * **[Acoustic rendering]** Sound propagation can be modeled as a linear time-invariant (LTI) system. In such a system, **all propagation paths and reflections** are captured by the impulse response [4]. To render the sound at an arbitrary location, we convolve the predicted impulse with the original signal in the time domain. Our approach is fully differentiable.
>
> In the revision, we will further clarify the differences between our model and NeRF.
>
> > **Q2) Memorization and network generalization.**
>
>
> * **[Memorization performs poorly]** We present **unambiguous evidence** that our model performance cannot be explained by memorization. We compare against strong nearest-neighbor and linear interpolation baselines in our paper at unseen locations. During testing, these methods require magnitudes more storage at inference (tens of gigabytes compared to tens of megabytes for our model). Despite this test time data advantage, these interpolation baselines perform worse than our method.
> * **[Contribution of our architecture]** We also perform ablation studies in Table 1. and Figure 5., and demonstrate that alternative architectures which do not share or do not use geometric features have lower generalization performance. These experiment that our architecture meaningfully contributes to the generalization performance, as all methods are trained on the same dataset.
> In the revision, we will highlight the novelty of our work, and the advantage our approach has over strong interpolation baselines.
>
> > **Q3) Implicit representation versus dataset.**
>
> An implicit representation allows us to infer the sound for emitters and listeners placed at dense locations not present in the training data, and is very compact compared to using the full dataset.
>
> The advantage of using our model over the dataset is multifold:
> * **[Compactness]** The model is much more compact than the dataset. The model is only tens of megabytes compared to tens of gigabytes of the dataset.
> * **[Continuous representation]** The dataset is not densely defined, and we wish to infer the acoustic response at locations not in the dataset. By learning an implicit function, our network utilizes learns the properties of sound propagation, and can generalize better than interpolation baselines.
> * **[Joint visual learning]** Our model helps with downstream tasks that can not be easily achieved with the original dataset.
>
> We want to highlight the qualitative results on our **website**: https://sites.google.com/view/nafs-iclr-2022
>
> > **Q4) Grid features provided to the network.**
>
> The reviewer may have misunderstood our approach.
>
> We **do not** provide grid features from the dataset directly to the network. Our grid features are learned from the impulse response itself. During training, the network is only provided with emitter/listener location and orientation, and is supervised with the log-STFT of impulse response.
>
> > **Q5) Notation of v representing impulse.**
>
> We clarify that we learn the log-magnitude STFT at specific time/phase indices for our NAF. We will update our notation in the revision.
>
> > **Q6) Additional visualizations.**
>
> As shown in Figure A4., the original dataset is sparse. We will provide additional loudness visualizations using nearest neighbor and linear interpolation in an upcoming version of our paper.

---

> > ### Author Response · Authors · 2021-11-14
> > **Response to Reviewer HhWA (Part 2/2)**
> >
> > > **Q7) Parameterization of the local feature grid.**
> >
> > We included details on our local grid in section B of the appendix. We will further describe the details here.
> >
> > Briefly, we initialize a grid with 32x32 resolution and 64 feature maps. The grid elements are initialized i.i.d. from a normal distribution with 0 mean and 1/sqrt(64) variance. We normalize a scene coordinate to between [-1, 1] using the axis aligned bounding box of the scene. We then query the grid using bilinear sampling. The sampling process is differentiable. The grid is learned using STFT magnitude supervision.
> >
> > We will provide a revision with these included details.
> >
> > > **Q8) Notation and details.**
> >
> > We clarify that $\theta$ represents head angle, while $k$ is a binary variable that represents the left/right ear. We will include these details in the revision. We will also include the dimension of each variable in the revision.
> >
> > > **Q9) RGB only baseline.**
> >
> > We utilize a standard two stage NeRF (hierarchical sampling). Both networks take as input a scene coordinate, and outputs an RGB value and an density parameter. The training dataset consists of view matrices and RGB image pairs.
> >
> > To ensure fairness with the NAF + NeRF experiment, we augment the "fine" network in NeRF with a learnable grid. We utilize 64 coarse samples and 128 fine samples per ray, and sample 1024 rays per batch. Supervision is provided via photometric (MSE) loss.
> >
> > > **Q10) Joint training of NAF and NeRF.**
> >
> > For the joint training baseline, in addition to the two NeRF networks, we also jointly learn a NAF. The NAF maps listener/emitter coordinates and orientations to the log-STFT magnitude.
> >
> > We condition the NAF network on the same grid as the second stage NeRF network. During training, the loss consists of both photometric NeRF loss and the MSE spectrogram loss. The grid in this setup is jointly optimized by both visual and acoustic losses.
> >
> > [1] Mildenhall, Ben, et al. "Nerf: Representing scenes as neural radiance fields for view synthesis." ECCV (2020).
> >
> > [2] Sitzmann, Vincent, Michael Zollhöfer, and Gordon Wetzstein. "Scene representation networks: Continuous 3d-structure-aware neural scene representations." NeurIPS (2019).
> >
> > [3] Raghuvanshi, Nikunj, and John Snyder. "Parametric directional coding for precomputed sound propagation." SIGGRAPH (2018).
> >
> > [4] Pierce, Allan D. Acoustics: an introduction to its physical principles and applications. Springer, 2019.

---

> > > ### Comment · Reviewer_HhWA · 2021-11-28
> > > **Thanks for the detailed rebuttal!**
> > >
> > > Thank the reviewers for answering many of my questions in detail and it addressed some of my concerns. This paper presents an interesting problem, but some of the contributions are overclaimed (as mentioned in my review). The paper needs to be significantly improved in terms of both writing and experiments/results. I appreciate the many new results added in the revision stage, but I also agree with other Reviewers some of the changes in the rebuttal/revision deviate from the original paper. I would suggest the authors takes all these suggestions and polish these changes for a resubmission.
> > >
> > > Again, I thank the authors for taking the time for writing such a detailed rebuttal. I have updated the score accordingly.

---

> > > > ### Author Response · Authors · 2021-11-29
> > > > **Thank you for your feedback and hoping for a more positive evaluation!**
> > > >
> > > > Dear Reviewer HhWA,
> > > >
> > > > We deeply appreciate your feedback, and are grateful that you consider our work to be interesting. Our goal has never changed from our initial revision, and that is to learn a continuous representation of spatial acoustics from sparse training samples.
> > > >
> > > > The link we provided in the original paper included many qualitative results for our work, which we have further augmented with a direct comparison between our network and interpolation baselines: [https://sites.google.com/view/nafs-iclr-2022](https://sites.google.com/view/nafs-iclr-2022/home)
> > > >
> > > > As we show on the website, interpolation using the dataset alone cannot yield satisfactory results.
> > > >
> > > > We respectfully push back on the concerns of "overclaim". We **never** claim we can generalize to unseen scenes. In the context of neural implicit representations, usually the goal is to learn a representation that can generalize to unseen locations/views for a *single scene*, after training from sparse views from the *same scene*. In the absence of multiview consistency used in vision, we demonstrate that learning local spatial features is an applicable alternative for spatial acoustics.
> > > >
> > > > In our very first draft, we included a website with qualitative results where we demonstrate unequivocally that our networks can continuously infer the spatial audio at unseen locations. We have further offered the clarifications and addressed the concerns as requested, and have updated the revision accordingly. As the first work in modeling the continuous spatial audio in a scene, we hope to make this framework sufficiently general for public audiences and inspire follow-up work to explore more in this new emerging field.
> > > >
> > > > We hope you can consider a more positive evaluation of our work.
> > > >
> > > > Thank you for your time!
> > > >
> > > >
> > > > Best,
> > > >
> > > > Authors

---

> ### Author Response · Authors · 2021-11-26
> **Sincerely Look Forward to Your Feedback!**
>
> Dear Reviewer HhWA,
>
> Thank you again for your feedback. As the deadline for discussion is approaching, we would be happy to provide any additional clarifications that you may need.
>
> In our previous response, we have carefully studied your comments and made updates to the revision as summarized below:
> * Provided a discussion how we enable generalization in the absence of multiview consistency via local geometric features.
> * Added a discussion on the advantage of using an implicit function compared to using the dataset.
> * Clarified in the paper that the grid features are learned, and provided additional details on how we initialize the grid.
> * Modified our notation to distinguish the original impulse response, and the STFT representation of the impulse response at a specific magnitude/phase index.
> * Added a visualization using nearest-neighbor and linear interpolation baselines.
> * Provided a description of our variables used in each equation, and the dimension of each variable.
> * Clarified the objective used in our NeRF only training, and the objective used when training NeRF and NAF jointly.
>
> Please let us know if you have any questions remaining. We would be happy to do anything that we can that would be helpful in the time remaining!
>
> Thank you for your time!
>
> Best,
>
> Authors

---

### Official Review · Reviewer_V59y · 2021-11-02

**Correctness:** 4
**Technical Novelty And Significance:** 2
**Empirical Novelty And Significance:** 3
**Recommendation:** 3
**Confidence:** 4

**Main Review:**

Strengths:  This paper shows a concrete methodology on how one can learn a NAF.  The use of a spectrotemporal representation, the particular way of representing positional information, the use of a local feature grid, and the architecture used, are interesting to know for anyone wanting to replicate that work.

Weaknesses: There are a lot of questions that arise in this work, and I was disappointed to see space being used for unnecessary fluff (wave equation?!!?) as opposed to a deeper investigation.

-- The choice to use a spectrotemporal representation is interesting, but it isn't clear why it works.  If the goal is to represent an RIR, a spectrogram would take at least as many values at the time-domain representation (actually more if you use windowing, which one should).  Why it is that it works better than using the time-domain data?  It isn't clear from the writing, but I assume that by "log spectrogram" you mean log( abs( STFT)) and not log( STFT).  If this is the case then you are not learning a room RIR, but something far simpler, which might explain why the problem becomes more tractable.  Is this the case?  If so, why would anyone care to estimate that quantity?  It's certainly of no value to any realistic audio problem, and doesn't resolve the issue of estimating the RIR which is what you start with.  Linked to this, comes the question of the loss function.  What is the MSE of the log spectrogram of an RIR, and why would anyone care to optimize it?  Clearly a lot of the most useful RIR information is in the phase component.

-- The experiments are really weak.  I think the most solid experimental point is figure 5, which conclusively supports the use of geometry conditioning.  But the experiments in sections 4.4 and 4.5 seem very contrived.  Of course the NAFs will help the NeRFs since they are smoother functions (due to ignoring phase), and the localization task used is just unrealistic.  I don't see how these experiments strengthen any point in this paper, they are simple more elaborate variations of your MSE estimate.

Minor things: "in principal" ⟶ "in principle"

**Summary Of The Paper:**

This paper makes use of the NeRF idea on RIRs (coined here as NAFs).  The main idea is relatively straightforward, sample a room from multiple emitter/receiver locations and learn to predict the resulting RIR using a neural net that receives as input only the involved locations, and some positional information.  It is shown that this approach learns something, and it is used in a few demonstrative examples such as sound localization and enhancing NeRF estimation.

**Summary Of The Review:**

Just as one can learn NeRFs, the authors show that it is possible to also learn (some version) of RIRs.  Representing RIRs is a big deal, and this paper can potentially be the first one published to use this approach.  However, had this been submitted to an audio conference I predict that it would be summarily rejected.  Unless I misunderstood your representation of the RIR, the problem solved is one that is made to be solvable, but not particularly useful for any real problem.  The log spectrogram of an RIR is not a quantity that one would employ in any audio processing, and the degree of approximation shown in figure 3 is way off from a high fidelity result (alas there were no qualitative listening experiments in this paper).  Although I appreciate the initiative, I would have expected a lot more depth in the first neural RIR paper, and this is coming short by a lot.  It's nice to see that a simpler version of this problem is solvable, but I do not see a useful contribution here.

If, on the other hand, I'm mistaken and you are actually estimating the phase as well, then I would like to see some results with more realistic audio experiments and potentially some quantitative listening numbers.

---

> ### Author Response · Authors · 2021-11-14
> **Response to Reviewer V59y (Part 1/2)**
>
> We are strongly encouraged by your evaluation that our work is the first on this topic, and could potentially be a big deal. We address specific comments below and **refer to the general response** for results. We will update the paper with the additional experiments and details.
>
> First, we want to note that we already provided qualitative audio results on the first page of our paper: https://sites.google.com/view/nafs-iclr-2022
>
> We observe that our system can continuously model the directionality, loudness decay, and portaling effects of doors as an agent moves in a scene. We will provide additional qualitative results that utilize linear and nearest neighbor baselines in an upcoming revision.
>
> We will highlight the link in an upcoming revision.
>
> > **Q1) Why use a STFT representation?**
>
> * **[Time domain modeling]** We acknowledge that the most straightforward approach would be direct modeling of the signal in the time domain. However, in initial experiments using the time domain, the models struggled to converge. Other papers using implicit networks for audio representations have also decided to either model the log-magnitude of the STFT or the full magnitude-phase STFT [1,2].
> * **[Fourier encoding and implicit networks]** Empirically, we observed the maximum frequency modeled by the network is effectively upper-bounded by the maximum frequency of the fourier encoding when modeling in the time domain. However as demonstrated by [3], using higher frequencies in the encoding leads to noise in the representation. By factorizing the time domain information into a time-localized frequency dependent representation, we have a representation that is easier to model. The choice of modeling phase in STFT is task specific [4,5,6], and does not change the smoother nature of STFT compared to the time domain representation.
>
> We will include results on time domain modeling in an upcoming revision.
>
> > **Q2) Reconstruction of RIR using magnitude**
>
> This step indeed needs to be better described in our paper, and we thank the reviewer for pointing this out. We model log(abs(STFT)), which is the log of the real component of the STFT. We leverage the temporal redundancy inherent in a STFT to recover the phase:
>
> * From the magnitude only STFT representation, we use the widely used iterative Griffin-Lim to reconstruct the phase component.
> * Once we have the phase, we can compute the inverse-STFT to recover the time domain (wave) representation of the impulse response.
>
> Recovering the phase using Griffin-Lim when only modeling the magnitude is an approach that has been utilized in other recent implicit audio learning work [1], and non-implicit sound synthesis tasks [4,5].
>
> We want to note that other learned and traditional approaches for spatial audio often **do not model phase**. These approaches utilize random phase reconstruction with learned log-magnitude-STFT [7], random phase [8] or minimum phase [9] filters. In human hearing for *impulse response* modeling, the onset time provides more important spatial cues [10,11] for low frequencies (~2kHz), while for higher frequencies the phase is not perceptible by humans [12]. Because we model the STFT, our work can capture the onset time.
>
> We will include additional experiments for STFT+phase modeling in an upcoming revision of our paper.

---

> > ### Author Response · Authors · 2021-11-14
> > **Response to Reviewer V59y (Part 2/2)**
> >
> > > **Q3) Other evaluation metrics.**
> >
> > We agree that it is very valuable to have additional evaluation metrics.
> >
> > We want to note that MSE/L1 losses on the STFT magnitude have been used for evaluation and learning of sound generation [13,14]. Furthermore, the Griffin-Lim algorithm iteratively infers the phase from the magnitude by minimizing the MSE between inferred and provided real magnitude-STFT.
> >
> > To better characterize the performance of our models, we also present reverberation time (T60), a metric used to characterize the performance of impulse response modeling in other recent work [7].
> >
> > Please refer to section **2.1** of our general response for T60 error metrics of our methods.
> >
> > We will provide additional T60 measurements in an upcoming revision.
> >
> > > **Q4) Do NAFs help NeRF only because of the smooth representation?**
> >
> > We agree that low frequency regularization could be one possible factor in the visual results.
> >
> > To demonstrate that the improved visual results do not come solely from the low frequency regularization, we investigate if applying L2 regularization to encourage a smooth latent space will improve the visual quality at unseen locations. We set the lambda_reg for the grid to 1e-5, as it provides approximately same gradient magnitude from regularization as reconstruction. The result is presented in section **2.2** of the general response.
> >
> > This shows that the improved generalization capability does not come solely from a smooth grid latent, but that our NAFs learn to encode spatially relevant information.
> >
> > We will provide additional experiments on linear-probe decoding in an upcoming version of our paper.
> >
> > We deeply appreciate the feedback from you, and will incorporate your suggestions in an upcoming revision of our paper. Please let us know if you have any other questions.
> >
> > [1] Du, Yilun, et al. "Learning Signal-Agnostic Implicit Manifolds." NeurIPS (2021).
> >
> > [2] Gao, Ruohan, et al. "ObjectFolder: A Dataset of Objects with Implicit Visual, Auditory, and Tactile Representations." CoRL (2021).
> >
> > [3] Tancik, Matthew, et al. "Fourier features let networks learn high frequency functions in low dimensional domains." NeurIPS (2020).
> >
> > [4] Wang, Yuxuan, et al. "Tacotron: Towards end-to-end speech synthesis." Interspeech (2017).
> >
> > [5] Ren, Yi, et al. "Almost unsupervised text to speech and automatic speech recognition." ICML (2019).
> >
> > [6] Engel, Jesse, et al. "Gansynth: Adversarial neural audio synthesis." ICLR (2019).
> >
> > [7] Singh, Nikhil, et al. "Image2Reverb: Cross-Modal Reverb Impulse Response Synthesis." ICCV (2021).
> >
> > [8] Pulkki, Ville. "Applications of directional audio coding in audio." ICA 2007.
> >
> > [9] Raghuvanshi, Nikunj, and John Snyder. "Parametric directional coding for precomputed sound propagation." SIGGRAPH (2018).
> >
> > [10] Chaitanya, Chakravarty R. Alla, et al. "Directional sources and listeners in interactive sound propagation using reciprocal wave field coding." SIGGRAPH (2020).
> >
> > [11] Shinn-Cunningham, Barbara G., Scott Santarelli, and Norbert Kopco. "Tori of confusion: Binaural localization cues for sources within reach of a listener." JASA (2000).
> >
> > [12] Oxenham, Andrew J. "How we hear: The perception and neural coding of sound." Annual review of psychology (2018).
> >
> > [13] Shen, Jonathan, et al. "Natural tts synthesis by conditioning wavenet on mel spectrogram predictions." ICASSP (2018).
> >
> > [14] Défossez, Alexandre, et al. "Sing: Symbol-to-instrument neural generator." NeurIPS (2018).

---

> > ### Comment · Reviewer_V59y · 2021-11-22
> > **Issues with moving from magnitude spectrogram to time domain**
> >
> > I am not quite comfortable with the proposal above.  Griffin-Lim was developed to work with specific types of signals, which didn't include RIRs.  At the simplest possible case in your problem you would have a set of ITD filters which means that a couple of STFT frames in each channel would have a flat magnitude spectrum.  Griffin-Lim cannot accurately estimate the location of the original deltas in the time domain (try it), this is an ill-posed ambiguous problem that cannot be solved this way.  I would be happy to be shown that I'm wrong here, but this comment is even supported by your plots in appendix F where the early reflections are not properly reconstructed.
> >
> > If you can quantitatively show me that you can accurately reconstruct the RIRs in the time domain (e.g. an MSE of the time domain RIR) I would accept that you indeed model the underlying responses as you claim in introduction.  But as it stands you are proposing one thing, but delivering something else.  In fact, I would be more positive if you modeled the phase instead of the magnitudes (which would at least keep the all important timing of the early reflections intact).
> >
> > As is I find the evaluation of the results very obfuscated, and I do not believe that the original claims are being addressed.

---

> > > ### Author Response · Authors · 2021-11-23
> > > **Response to V59y (magnitude to time)**
> > >
> > > Thank you for the prompt response!
> > >
> > > We agree that phase is very important for a variety of tasks (including music synthesis and microphone-array direction estimation), and the phase recovered by Griffin-Lim may not be sufficiently faithful for the spatial impulse response in these applications. However, there are two reasons that we think using a neural implicit function to learn the magnitude alone is useful and meaningful.
> > >
> > > - [**Qualitative results are good enough for game and VR applications**] For many widespread uses of spatial audio (gaming/VR), the goal is to provide a plausible representation of sound propagation and reverberation in a scene. For these use cases, the phase is not necessarily critical to providing a believable experience, and is often not modeled (notably Raghuvanshi's parametric coding, Drori's image2reverb). As we show on our website, we can still present a plausible spatial audio experience at novel continuous locations after trained on only sparse samples, while using a magnitude less storage compared to the original dataset.
> > >
> > > - [**Phase is hard to learn using neural implicit functions**] Following your suggestion, we did experiment with learning phase using an implicit function, however the naturally smooth nature of implicit functions and the spatially chaotic nature of phase meant that the network struggled to accurately generalize phase at unseen locations using the current parameter count. Attempting to learn the phase as an additional output to the current network led to degraded T60 and STFT MSE metrics. We speculate the reason is that phase changes too quickly as a function of spatial location, and is difficult to learn from sparse samples of a scene. We also added results that directly regress the waveform using an implicit function, however this approach struggles to model higher frequencies. We believe that as the first work in this field, our method of modeling magnitude is a valid initial step towards using a neural field to learn spatial reverberations.
> > >
> > > - [**Future work: NAF + neural vocoder**] We believe that a potential avenue for exploration would be to combine our implicit NAF STFT output with a principled neural vocoder (like Wavenet). While there are many possible approaches for recovering the phase, we believe they are orthogonal to the scope of our task. We are happy to clarify our goals more clearly, include a discussion of the limitations of our work, and describe possible avenues for future work in neural waveform recovery.
> > >
> > > Thanks again for your insightful comments. We have gained much from your constructive suggestions, and have made a good faith effort to address your concerns in our revisions. We hope that we have been able to clarify our goals and limitations of this work. We believe this is the first study that tackles learning plausible spatial audio using a neural field. Indeed, completing this first step has not been trivial; more importantly, it lays the foundation for future exciting improvements that can fully address your suggestions. We hope that as an expert in the field, you will support our efforts of expanding the field and consider a more positive evaluation given to our work.
> > >
> > > We are happy to answer any additional questions.

---

> > > > ### Comment · Reviewer_V59y · 2021-11-26
> > > > **Response to authors**
> > > >
> > > > I certainly understand your point of view and I appreciate the difficulty of the task.  But we started from "We demonstrate that NAFs capture environment reverberations of a scene with high fidelity" and we have now moved to "qualitative results are good enough for games" and "phase is hard to learn".  That's ok, but this is quite a deviation from the original paper, and I would have expected to see different experiments and benchmarks to support these new statements.  Can you demonstrate that it's good for games/VR?  Is it in any way better than simpler models?  If you will not be modeling phase, then a lot of your references are not directly relevant, you should be looking at comparisons with work focusing on reverb coloration and room EQing.
> > > >
> > > > I think this is good work, but there's a disappointing disconnect between the promise, the deliverables, and the evaluation.

---

> > > > > ### Author Response · Authors · 2021-11-28
> > > > > **Response to V59y**
> > > > >
> > > > > Thank you for your comments. We have worked hard to address your concerns, and we have incorporated your suggestions in an revision. We are happy to address any remaining concerns with additional clarifications followed by paper revision.
> > > > >
> > > > > > **Comparison with simple baselines.**
> > > > >
> > > > > We have provided a comparison against two simple but strong baselines (nearest neighbor and linear interpolation) and show quantitative results in Table 1., with qualitative results shown in Figure A5 of the Appendix. We demonstrate that we achieve lower error when measured in magnitude-STFT, and achieve lower average error when measured using T60 as a metric. We are currently working on additional qualitative results for our website, this will take one or two days.
> > > > >
> > > > > > **Advantages of our NAFs over interpolation baselines.**
> > > > >
> > > > > A further important advantage is the compact nature of the implicit representation. On average our NAFs use 0.5% the storage of these interpolation baseline methods as we show in Table A1 of the Appendix. The compact nature of the implicit representation means that we can encode and utilize spatial audio even when we cannot store the full amount of spatial audio data (<30MB for NAFs, several GBs for interpolation).
> > > > >
> > > > > > **Limitations and future work.**
> > > > >
> > > > > In terms of only modeling the magnitude of the STFT for a spatial impulse response, precedent can be found in the Image2Reverb (ICCV 2021) paper. They describe modeling the magnitude-only STFT (and sample random phase) as a way to model the spatial impulse response in high quality.
> > > > >
> > > > > We agree that modeling the magnitude alone cannot account for all the information in the impulse response. We will clarify our goal and the limitations of our work in an upcoming revision.
> > > > >
> > > > > > **Consistency on paper writing.**
> > > > >
> > > > > As the first work in modeling the continuous spatial audio in a scene, we strive to make this framework general to public audiences and inspire follow-up work to explore more in this new emerging field. We understand your concerns, and are happy and able to make the change in the text to make our application and claim more appropriate.
> > > > >
> > > > > Because we are no longer able to update the original paper, we **provide an updated revision on [our website](https://sites.google.com/view/nafs-iclr-2022/home).** We have modified our language in the introduction to reflect our goal is to model plausible spatial audio, We have also modified our conclusion to discuss the limitations of our current model, and potential avenues for future work.
> > > > >
> > > > > Please feel free to let us know if you have additional suggestions!

---

> > > > > ### Author Response · Authors · 2021-11-29
> > > > > **Website updated with direct comparison**
> > > > >
> > > > > We have updated our website with an revision of our paper, and new qualitative results using nearest neighbor and linear interpolation baselines.  In both baselines, we perform interpolation directly in the time domain with the original waveform. We restrict our interpolation to the same dataset as used for training the NAF. The original training dataset is roughly 10 GB in size, while our NAF is significantly more compact at less than 30 MB.
> > > > >
> > > > > [https://sites.google.com/view/nafs-iclr-2022/home](https://sites.google.com/view/nafs-iclr-2022/home)
> > > > >
> > > > > * For the linear interpolation approach, note how the energy on the opposite side of the wall incorrectly influences the sound hear on the other side, this causes the audio to sound very close even when we are separated by a wall. Also note that as we exit the convex hull defined by the training data, we cannot correctly infer the spatial audio.
> > > > > * For the nearest neighbor approach, note the sudden loudness change at 0:28. When the underlying data is sparse, this approach can jump suddenly from one impulse response to another, leading to jarring changes in the spatial audio.
> > > > >
> > > > > In comparison, our approach can continuously infer a plausible spatial audio at various locations.

---

### Official Review · Reviewer_doFB · 2021-11-02

**Correctness:** 3
**Technical Novelty And Significance:** 4
**Empirical Novelty And Significance:** 3
**Recommendation:** 6
**Confidence:** 4

**Main Review:**

Strengths:
The proposed idea is novel and expected to promote other related researches in the future.

Weaknesses:
I think the details written in this paper is not enough to reproduce the results.
1. How can one listen to the predicted impulse response when the estimate values are only the magnitude parts?
2. How were the visualization done in Figure 1 and 4?
3. What happens on the local grid parts where the impulse response was not collected enough if not at all? I assume those parts will not be trained and results in poor prediction. Is this correct?
4. in section 4.2, the authors say something like "processed with sinusoidal encoding", but the details are not enough.

**Summary Of The Paper:**

This paper proposes an neural acoustic field (NAF), which is an extended version of neural radience field applied on impulse responses defined between listener and emitter positions. It predicts an impulse response given the listener and emitter positions so that one can simulate the impulse response in arbitrary positions. One of the key idea to extend the model to unseen positions is to take local grid features as input. The authors show other applications with NAF such as source localization and multi-modal NERF. Experiment results show that the proposed method shows better results when comparing to baseline methods.

**Summary Of The Review:**

The idea is seemingly novel but it is not enough for publication in this status.
I recommend reject. But I'm willing to raise the score if the authors answer the questions with more details and write them in the paper.

---

> ### Author Response · Authors · 2021-11-14
> **Response to Reviewer doFB**
>
> We thank Reviewer doFB for the detailed and constructive review. We address specific comments below, and will include additional details in a revision.
>
> > **Q1) How one can listen to the predicted impulse response when the estimate values are only the magnitude parts.**
>
>
> We provided qualitative audio results in this website included in the paper, reproduced below: https://sites.google.com/view/nafs-iclr-2022
>
> We demonstrate that our model can continuously capture the acoustic effects (reverberation, directionality, portaling due to doors) of a 3D scene after training on sparse data.
>
> Prior works on spatial impulse response modeling also do not model phase information. During the synthesis stage - learned approaches [1], fixed position approaches [2], and approaches that allow for listener/emitter movement [3] respectively use either random phase with magnitude-STFT of the impulse, or random phase/minimum phase filters respectively.
>
>
> To compute the time domain impulse response from magnitude only, we can utilize the redundancy present in the STFT representation to recover the phase:
>
> 1. Given the magnitude only STFT representation, we apply the widely used iterative Griffin-Lim [4] algorithm to reconstruct the phase. After reconstructing the phase, we can perform inverse STFT to derive the time domain (wave) representation of the impulse response.
> 2. After computing the time domain impulse response, we convolve with an audio sample to render the final result.
>
> This approach of modeling magnitude-only STFT and inferring phase with Griffin-Lim is also used in other recent work that learn audio generation [5, 6].
>
> We will include these details in an upcoming revision of the paper.
>
> > **Q2) How the visualizations are done in Figure 1 and 4.**
>
> For a given emitter position, we iterate over all positions in a scene for the listener and infer the STFT of the impulse response. Given this real magnitude component, we take the sum across all frequency bands up to a maximum time limit, and take the log to improve contrast for visualization. This approach is also used in the soundspaces paper for their loudness visualization [7].
>
> > **Q3) Performance on unobserved locations?**
>
> We agree that the performance of NAF on unobserved speaker/emitter locations is important.
>
> * **[Evaluation on unobserved locations]** We would like to clarify that, in fact, our evaluation in Table 1. and Figure 5. is performed precisely on unseen combinations of emitter/listener locations. This demonstrates that our approach of leveraging a learned spatial representation generalizes better compared to interpolation baselines and MLP approaches even when provided with very sparse training data.
> * **[Generalization with learnable geometric attributes]** As is true for implicit visual learning approaches as well (NeRF, SRN, etc.), our model does requires *some* observations to be able to operate in a room. In vision, we can leverage multiview consistency and learn a dense representation using photometric loss. However, we cannot assume multiview consistency in sound propagation. Instead we leverage the fact that geometry local to the listener/emitter strongly influence the anisotropic sound propagation [8]. Our framework learns a spatial representation which captures the local attributes. This in turn allows us to generalize to continuous locations from sparse training data. The learned features are visualized in Figure 6.
>
> > **Q4) Details of the sinusoidal encoding.**
>
> We note that in our initial submission, we did include details on the sinusoidal encoding on page 11 in the appendix. We can move this information to the main paper.
>
> We utilize 10 frequencies linearly spaced between 1 Hz and 100Hz/200Hz for position/time-frequency respectively. We did not observe performance improvements from further increasing the maximum frequency. Both sine and cosine functions were used following NeRF.
>
> > **Q5) Details for reproducibility.**
>
> We will provide a copy of the code in the review period.
>
> We sincerely appreciate your comments. Please feel free to let us know if you have further questions.
>
> [1] Singh, Nikhil, et al. "Image2Reverb: Cross-Modal Reverb Impulse Response Synthesis." ICCV (2021).
>
> [2] Pulkki, Ville. "Applications of directional audio coding in audio." ICA 2007.
>
> [3] Raghuvanshi, Nikunj, and John Snyder. "Parametric directional coding for precomputed sound propagation." SIGGRAPH (2018).
>
> [4] Griffin, Daniel, and Jae Lim. "Signal estimation from modified short-time Fourier transform." IEEE ASSP (1984).
>
> [5] Du, Yilun, et al. "Learning Signal-Agnostic Implicit Manifolds." NeurIPS (2021).
>
> [6] Wang, Yuxuan, et al. "Tacotron: Towards end-to-end speech synthesis." Interspeech (2017).
>
> [7] Chen, Changan, et al. "Soundspaces: Audio-visual navigation in 3d environments." ECCV (2020).
>
> [8] Raghuvanshi, Nikunj, and John Snyder. "Parametric directional coding for precomputed sound propagation." SIGGRAPH (2018).

---

> > ### Comment · Reviewer_doFB · 2021-11-23
> > **Raising score to 6**
> >
> > The authors have answered all my questions in detail.
> > Although there are still some concerns left, I believe this study could be a nice first step of nerf being extended to spatial sound.
> > I thank authors for their efforts to answer the questions of reviewers.

---

### Author Response · Authors · 2021-11-14
**[Pre-revision] General response (Part 1/3)**

We thank all reviewers for their constructive comments.

We are very encouraged by reviewers' evaluation on the novelty and significance of this work. All four reviewers find that our work on Neural Acoustic Fields (NAFs) is novel and interesting ("proposed idea is novel" (doFB), "is new and interesting" (HhWA), "very interesting" (U1r6), "representing RIRs is a big deal" (V59y)).

All reviewers pointed out that we should better present the work. We totally agree. Indeed, we now note some obvious items that we can immediately improve. For example, although we have provided a demo to illustrate the qualitative results, we should have highlighted the link so that the reviewers can have an intuitive experience of our work.

We will provide additional qualitative and quantitative results in an upcoming revision. By adding some new data, providing more detailed descriptions, and clarifying the key issues, the work will be substantially improved, and **we are very confident that all reviewers' concerns will be fully addressed**. First, we would like to make several clarifications on the common concerns.

## 1. General Clarification
### 1.1 Qualitative results
Several reviewers asked for demos of qualitative results. We actually provided a link in the original submission of our paper to the demos, which we list below again:
https://sites.google.com/view/nafs-iclr-2022.

These demos are important because they show that, after training using sparse data, we can continuously infer the sound at arbitrary locations and orientations in a scene. Therefore, they answer reviewers' questions if our approach can learn realistic acoustic effects including reverberation, decay, and portaling effects as the listener traverses the scene.

In our revision we will highlight the link.

### 1.2 Scope and goal of our work
Reviewers recommended that we be explicit about the scope and goal of this study. Here, we study how to represent sound propagation in an individual scene as a continuous implicit function.
* We are the first to propose learning the complete continuous sound field of a scene. Our approach uses only emitter/listener position & orientation/left-right as input, and is supervised with sparse impulse responses.
* We present an architecture that improves the generalization capability when learning sound propagation in lieu of a vision like photometric loss
* Given this learned representation, we demonstrate that we can render the propagated sound at continuous locations in a scene for arbitrarily positioned emitters (see **section 1.1**).
* We further demonstrate that our NAFs learn a representation useful for visual rendering as well.


### 1.3 Memorization and interpolation
Our initial submission led to confusion among two reviewers that our model achieved the performance by mere memorization. This is not the case at all.

In vision based models (NeRF, SRN, etc.), generalization to unobserved viewpoints is achieved by enforcing a multiview consistent constraint via photometric loss [1, 2]. However the same assumptions cannot be made in sound propagation. Instead because the anisotropic reflections are strongly affected by geometry close to the listener and emitter [3], we propose to learn the necessary local geometric properties of a scene.

In our paper we include strong nearest-neighbor and linear interpolation baselines. Despite requiring magnitudes more storage at inference (up to tens of gigabytes compared to tens of megabytes for our method), these interpolation-based memorization approaches perform significantly worse when evaluated at unobserved locations. Furthermore, the nearest-neighbor baseline does not allow for smooth changes when provided with sparse training samples, while linear baselines are well understood to have audible artifacts [4].


Our qualitative results provided in **1.1** demonstrate that our proposed framework can capture the acoustic effects at continuous locations in the scene.

---

> ### Author Response · Authors · 2021-11-14
> **[Pre-revision] General response (Part 2/3)**
>
> ### 1.4 Learning a magnitude only STFT representation
>
> Three reviewers asked us to clarify how the time domain signal can be recovered given that we only learn the magnitude of the STFT. This is indeed not very straightforward. We will provide a much better clarification on how acoustic rendering can be achieved using our network.
>
> * Because the Short-time Fourier transform (STFT) is computed with overlapping windows, the phase information can be recovered by using the redundancy of the magnitude representation.
> * In our paper we employ the widely used iterative Griffin-Lim algorithm [5] to reconstruct the phase from the magnitude of the STFT, and infer the time domain impulse response.
> * Once we infer the impulse, we render the sound by convolving the source signal utilizing convolution.
>
> Most prior works on spatial impulse response modeling do not model phase information. During the synthesis stage - learned approaches [6], fixed position approaches [7], and approaches that allow for listener/emitter movement [3] respectively use either random phase with magnitude-STFT of the impulse, random phase filters, or minimum phase filters respectively. In the context of spatial impulse responses, for lower frequencies (below ~2kHz) the time delay provides more important spatial cues [8,9], while at higher frequencies the phase is generally not perceptible [10]. We capture the time delay by utilizing the STFT representation, which captures time-localized frequency information.
>
> We will update our paper to include these details. Our framework is also compatible with jointly learning phase and magnitude, and we will provide results in a revised version of our paper.
>
> ## 2. Additional Results (pre-revision)
> ### 2.1 Time domain metric of our methods
> Reviewer V59y asked whether MSE on the STFT representation alone is sufficiently informative. While we note that several past works on learned audio modeling have utilized MSE/L1 on STFT to train or evaluate their models [11, 12], we also agree that we can better measure the quality of the generated impulse responses.
>
> Here we present results using the reverberation time (T60) metric, which is a widely used metric for quantifying room impulse responses and has been utilized in prior work [13, 14]. We show the average percentage difference for T60 between generated and ground truth impulses responses.
>
> | Model                   | Large 1 | Large 2 | Medium 1 | Medium 2 | Small 1 | Small 2 | Mean |
> |-------------------------|---------|---------|----------|----------|---------|---------|------|
> | NAF (Dual local feat)   | 4.31    | 4.88    | 5.00     | 4.20     | 6.68    | 3.80    | 4.81 |
> | NAF (Shared local feat) | 4.16    | 5.24    | 3.80     | 3.88     | 6.75    | 3.77    | 4.60 |
>
> ### 2.2 Regularization of the NeRF latent
> Two reviewers asked us to provide details on the NAF+NeRF joint learning experiment. Here we present results where an L2 penalty with weight 1e-5 is applied to the NeRF grid to encourage a smooth latent under RGB only supervision. In this experiment we train with 75 images.
>
> | Large 1 | PSNR | MSE (1e-3) | Large 2 | PSNR | MSE (1e-3) |
> |---|---|---|---|---|---|
> | NeRF + L2 reg| 22.69 | 6.956 | NeRF + L2 reg| 24.86 | 7.128 |
> | NeRF  | 25.41 | 6.618 | NeRF  | 25.70 | 6.921 |
>
>
> ## 3. Planned Revisions
> ### 3.1 Additional experiments and clarification
> 1. We will add a time domain metric most often used to characterize room impulse responses (T60).
> 2. We will implement a network to jointly learn the magnitude and phase of the impulse, and analyze their performance using spectrogram and time domain metrics.
> 3. We will add a linear-probe decoding experiment to demonstrate that the learned grid features contain spatially useful information.
> 4. We will add additional qualitative results for nearest-neighbor and linear impulse interpolation.
> 5. We will revise our paper carefully, including providing more technical details of our framework, giving specific information on how sound can be rendered from magnitude, describing model limitations, content adjustments, and grammar checking.

---

> > ### Author Response · Authors · 2021-11-14
> > **[Pre-revision] General response (Part 3/3)**
> >
> > ### 3.2 Reproducibility
> > We will provide a link to a copy of our code during the review period.
> >
> > ### Conclusion
> > We thank the reviewers for their helpful feedback and suggestions for additional evaluation, which will make the paper substantially stronger. We look forward to additional discussions.
> >
> > [1] Mildenhall, Ben, et al. "Nerf: Representing scenes as neural radiance fields for view synthesis." ECCV (2020).
> >
> > [2] Sitzmann, Vincent, Michael Zollhöfer, and Gordon Wetzstein. "Scene representation networks: Continuous 3d-structure-aware neural scene representations." NeurIPS (2019).
> >
> > [3] Raghuvanshi, Nikunj, and John Snyder. "Parametric directional coding for precomputed sound propagation." SIGGRAPH (2018).
> >
> > [4] Raghuvanshi, Nikunj, et al. "Precomputed wave simulation for real-time sound propagation of dynamic sources in complex scenes." SIGGRAPH (2010).
> >
> > [5] Griffin, Daniel, and Jae Lim. "Signal estimation from modified short-time Fourier transform." IEEE ASSP (1984).
> >
> > [6] Singh, Nikhil, et al. "Image2Reverb: Cross-Modal Reverb Impulse Response Synthesis." ICCV (2021).
> >
> > [7] Pulkki, Ville. "Applications of directional audio coding in audio." ICA 2007.
> >
> > [8] Chaitanya, Chakravarty R. Alla, et al. "Directional sources and listeners in interactive sound propagation using reciprocal wave field coding." SIGGRAPH (2020).
> >
> > [9] Shinn-Cunningham, Barbara G., Scott Santarelli, and Norbert Kopco. "Tori of confusion: Binaural localization cues for sources within reach of a listener." JASA (2000).
> >
> > [10] Oxenham, Andrew J. "How we hear: The perception and neural coding of sound." Annual review of psychology (2018).
> >
> > [11] Shen, Jonathan, et al. "Natural tts synthesis by conditioning wavenet on mel spectrogram predictions." ICASSP (2018).
> >
> > [12] Défossez, Alexandre, et al. "Sing: Symbol-to-instrument neural generator." NeurIPS (2018).
> >
> > [13] Singh, Nikhil, et al. "Image2Reverb: Cross-Modal Reverb Impulse Response Synthesis." ICCV (2021).
> >
> > [14] Tang, Zhenyu, et al. "Scene-aware audio rendering via deep acoustic analysis." TVCG (2020).

---

### Author Response · Authors · 2021-11-22
**Revision Updated (Part 1/2)**

We would like to thank all reviewers again for their constructive suggestions and feedback.

According to your comments, we have provided an updated revision with a detailed response to each reviewer's questions and concerns. We have revised our paper with the following changes:

## Changes to the abstract
* We move the link to our qualitative results so it is more visible. (Reviewer doFB, V59y, HhWA, U1r6)

## Changes to the introduction
* We provide additional details on the scope of our work, and of similar papers in the visual domain. We also clarify our goals. (Reviewer doFB, HhWA, U1r6)
* We clarify the advantage that a neural acoustic field has over using the dataset in the introduction. (Reviewer HhWA)
* We motivate our work, and why we want to learn local geometric features to help with generalization. (Reviewer doFB, HhWA, U1r6)
* We clarify how the visualization in Figure 1. is done in the caption. (Reviewer doFB)

## Changes to the related work
* We provide additional related work that learn to generate the phase-free magnitude STFT of sound. (Reviewer V59y)

## Changes to methods section
* We have compressed the description for wave equation background in section 3.1 significantly (Reviewer V59y)
* We add a paragraph describing how audio can be rendered, and why our/prior work choose not to model phase in Section 3.2. (Reviewer doFB, V59y, U1r6)
* We mention alternative learned approaches to recover the waveform in section 3.2 (Reviewer V59y)
* We clarify that the grid is learned and how it is queried in section 3.3 (Reviewer HhWA, U1r6)
* We have clarified our notation surrounding v, to indicate that our network model outputs $v_{\text{STFT}}$ for a given time/phase in Equation 4, 5, 6 in section 3.2 (Reviewer HhWA)
* We now explain that $\theta$ represents head orientation, and $k$ represents ear (left/right) before equation 3, and note the dimension of every input in section 3.2 (Reviewer HhWA, U1r6)
* We use different notation to distinguish the time domain impulse response function $\phi$, the network to approximate STFT as $\Omega$, and the NAF with the grid as $\Omega_\text{grid}$. (Reviewer U1r6)

## Changes to the experiments section
* We clarify that the test set is selected randomly in section 4.1 (Reviewer U1r6)
* We add additional clarification on our sinusoidal encoding in section 4.2. (Reviewer doFB)
* We add T60 percentage difference as a metric as in Image2Reverb in Table 1. (Reviewer V59y)
* We add a linear decoding experiment on the grid features in Section 4.4 results. (Reviewer V59y)
* We now note the dimension of every input, use $q$ and $q'$ to represent location, and use $x,y$ to represent the ground axis in section 4.2 (Reviewer HhWA, U1r6)
* We clarify that our NAF output the STFT log-magnitude at a specific time/frequency index (Reviewer HhWA)
* We add additional details for the NeRF baseline, including training objective and number of ray samples in section 4.4 (Reviewer HhWA)
* We add additional details on the loss used for the cross-modal training in section 4.4 (Reviewer HhWA)
* We clarify the max magnitude baseline, and note that equation 8 is used when we have a NAF used in section 4.5(Review U1r6)

## Changes to the appendix
* We provide additional details on how we perform zero-padding in Appendix B. (Reviewer U1r6)
* We add a comparison that trains to regress phase + log-magnitude, and one network that trains directly in the time domain waveform. Results are presented in the Table A2 and Figure A6 in Appendix E. (Reviewer V59y)
* We add an experiment that ensures a smooth latent space via a L2 penalty in Appendix G. (Reviewer V59y)
* We add a comparison that highlights the storage advantage our NAF over using the dataset in Appendix D. (Review HhWA)
* We provide a visualization using interpolation applied to the ground truth training data in Figure A5 in Appendix E (Reviewer HhWA)
* We provide more detail on how our local feature grid is initialized in Appendix B. (Reviewer HhWA, U1r6)
* We provide qualitative results for interpolating the "ear" variable in Appendix H (Reviewer U1r6)

---

> ### Author Response · Authors · 2021-11-22
> **Revision Updated (Part 2/2)**
>
> ## Code for reproducing results
> We provide a copy of our code [anonymously here](https://anonymous.4open.science/r/Neural_Acoustic_Fields/).
>
> ## Summary of new experiments/visualizations
> * We add T60 percentage error in Table 1. (Reviewer V59y)
> * We add a linear decoding experiment in section 4.4 (Reviewer V59y)
> * We compare modeling the log-magnitude, the log-magnitude + phase, and the time domain in Table A2 of the Appendix. (Reviewer V59y)
> * We visualize the waveform recovered from NAF using Griffin-Lim against a network trained directly in the time domain in Figure A6 of the Appendix. (Reviewer V59y)
> * We experiment with regularizing the grid presented to NeRF in Table A3 of the Appendix. (Reviewer V59y)
> * We add a comparison of the storage cost of the methods in Table A1 of the Appendix. (Reviewer HhWA)
> * We visualize the loudness maps when using either nearest or linear interpolation in Figure A5 of the Appendix. (Reviewer HhWA)
> * We visualize the effect of interpolating the "ear" latent in Figure A7 of the Appendix. (Reviewer U1r6)
>
> We hope our responses have convincingly addressed all reviewers' concerns. We thank all reviewers for their time and effort. Please do not hesitate to let us know of any additional comments or questions regarding the manuscript or the changes.

---

### Decision · Program_Chairs · 2022-01-20

**Decision:**

Reject

**Comment:**

This paper proposes to use implicit neural representations to model how our surroundings affect the sounds reverberating within. Concretely, the proposed approach can produce impulse responses that capture environment reverberations between any two points in a scene.

Reviewers praised the novelty and originality of the idea (and I concur), but raised concerns about the clarity of the writing (especially w.r.t. modelling the phase component), lack of detail, insufficient or inadequate experiments and overclaiming of results. (There were also concerns about overclaiming of contributions, but I am inclined to agree with the authors that this isn't really the case.)

The authors have clearly taken the time to try to address these concerns, and I commend them on their willingness to engage with the reviewers' comments and suggestions. While one of the reviewers raised their score to "accept", I am inclined to agree with the other reviewers and recommend rejection. The required degree of revision is substantial, and therefore difficult to assess within a single review cycle. I believe this work must undergo another thorough assessment in its revised form, before it can be accepted for publication.